

# Observation-based estimates of volume, heat and freshwater exchanges between the subpolar North Atlantic interior, its boundary currents and the atmosphere

Sam C. Jones[1], Neil J. Fraser[1], Stuart A. Cunningham[1], Alan D. Fox[1], Mark E. Inall[1]

[1]Scottish Association for Marine Science, Oban, UK

Correspondence to: Sam Jones (sam.jones@sams.ac.uk)

**Abstract**

The Atlantic Meridional Overturning Circulation (AMOC) transports heat and salt between the tropical Atlantic and Arctic oceans. The interior of the North Atlantic Subpolar Gyre (SPG) is responsible for the much of the
water mass transformation in the AMOC, and the export of this water to intensified boundary currents is crucial for projecting air-sea interaction onto the strength of the AMOC. However, the magnitude and location of exchange between the SPG and the boundary remains unclear.

We present a novel climatology of the SPG boundary using quality controlled CTD and Argo hydrography. We define the SPG as the oceanic region bounded by 47° N and the 1000m isobath. From this hydrography we
compute geostrophic currents referenced to altimetry across the SPG boundary.

Water density generally increases moving counter-clockwise from Biscay, leading to geostrophic flow out of the SPG around much of the boundary with minimal seasonality. An exception is the West Greenland Current region, where the density gradient is reversed, with a corresponding reversal in flow into the interior. Across the southern boundary at 47° N, geostrophic flow is generally into the SPG above 1000 m. In contrast the surface
Ekman forcing drives net flow out of the SPG in all seasons with pronounced seasonality, varying between 2.45 ± 0.73 Sv in the summer and 7.70 ± 2.90 Sv in the winter. We estimate heat advected into the SPG to be between 0.14 ± 0.05 PW in the winter and 0.23 ± 0.05 PW in the spring, and freshwater advected out of the SPG to be between 0.07 ± 0.02 Sv in the summer and 0.15 ± 0.02 Sv in the autumn. These estimates approximately balance the surface heat and freshwater fluxes over the SPG domain.

Overturning in the SPG varies seasonally, with a minimum of 6.20 ± 1.40 Sv in the autumn and a maximum of 10.17 ± 1.91 Sv in the spring. The primary density of maximum overturning is at 27.30 $kgm^{-3}$, with a secondary, smaller maximum at 27.54 $kgm^{-3}$. Upper waters ($\sigma_0 < 27.30$ $kgm^{-3}$) are transformed in the interior then exported as either intermediate water (27.30-27.54 $kgm^{-3}$) in the NAC or as dense water ($\sigma_0 > 27.54$ $kgm^{-3}$) exiting to the south. Our results support the present consensus that the formation and pre-conditioning of
subpolar Mode Water in the north-eastern Atlantic is a key driver and modulator of AMOC strength.

## 1. Introduction

The AMOC (Atlantic Meridional Overturning Circulation) is the zonally integrated system of currents transporting heat and salt between the tropics and high latitudes in the Atlantic Ocean. It is a key component of



the global thermohaline circulation, transporting approximately 25% of the global ocean-atmosphere heat

transport. Meridional heat transport associated with the AMOC is 1.2 PW across 26° N (RAPID, Smeed et al., 2018), diminishing to 0.51 PW at 58° N (OSNAP, Li et al., 2021b). The subpolar North Atlantic (SPNA) plays a large role in regulating the climate system by connecting surface and deep layers, such that variability in these regions can imprint on global averages and mediate the rate of climate change (Chen and Tung, 2014; IPCC 2021).

The SPNA features a cyclonic system of currents collectively termed the Subpolar Gyre (SPG), transporting warm, salty water northwards on its eastern side and transitioning into a cool, fresh southward flow on its western side. The strongest currents in the SPG are located around the periphery due mainly to meridional density gradients and topographic intensification in the east (Huthnance et al., 2022; Marsh, 2017), and western intensification in the west (Munk, 1950; Stommel, 1957; Sverdrup, 1947). The Gulf Stream is the primary input

of water to the SPG from the south. As the Gulf Stream crosses the Atlantic from west to east, a portion transitions into the North Atlantic Current (NAC); about 55 % of NAC transport is thought to circulate in the SPG while the remainder is diverted poleward over the Greenland-Scotland Ridge (Berx et al., 2013; Hansen et al., 2015; Østerhus et al., 2019). Return flow into the SPG from the Arctic and Nordic Seas mainly occurs in deep overflows over the Greenland-Scotland Ridge (Dickson et al., 2008; Johnson et al., 2017; Østerhus et al.,

50    2008).

Rather than forming a continuous current around the basin, the SPG rim plays host to numerous discrete current systems although drift trajectories and model particles can occasionally follow a continuous path around the gyre in boundary currents (Burrows et al., 1999; Fischer et al., 2018; Gary et al., 2020; Li et al., 2021a). Upper ocean currents on the eastern rim of the SPG are typically strongest near the surface and decrease with depth due

to pronounced stratification in the water column (Huthnance et al., 2020; Souza et al., 2001). With progress counter-clockwise around the SPG, stratification decreases and currents become increasingly barotropic (Hopkins et al., 2019; Pacini et al., 2020).

The canonical view of the AMOC functioning in the SPNA has been that winter-time buoyancy loss in the Labrador Sea drives deep convection, and that this convection was the principal direct linkage of the upper and

lower limbs of the overturning (e.g. Rhein et al., 2011). However, observations from the OSNAP array have transformed this view, providing strong evidence that the mean and variability in the SPG AMOC is driven by buoyancy exchanges in the ocean basins north of OSNAP-East (Irminger Basin, Iceland Basin and Rockall Trough) (Kostov et al., 2021, Li and Lozier 2018; Li et al., 2021a; Lozier et al., 2019; Petit et al., 2020; Petit et al., 2021). Processes north of the Greenland-Scotland Ridge (GSR) also contribute significantly to the mean

(Petit et al., 2021; Tsubouchi et al., 2021; Zhang and Thomas, 2021).

A reconciliation of these views is a new appreciation that most of density anomalies evident in the Labrador Sea are generated by buoyancy exchanges in the east and imported to the Labrador Sea. So, while they are an ultimate indicator of SPG AMOC functioning, they are not the source drivers (Li et al., 2021a; Menary et al., 2020). A remaining challenge for tracking the AMOC is therefore understanding the location, nature and

hierarchy of processes connecting this formation of light Subpolar Mode Water (SPMW, Brambilla and Talley, 2008; Brambilla et al., 2008) with the eventual export of dense waters in the lower limb.



One way of further refining our understanding of AMOC is to distinguish processes taking place in the SPG interior from those external to the SPG (mainly in the SPG boundary and north of the GSR, e.g. Desbruyères et al., 2020; Petit et al., 2021; Tsubouchi et al., 2021). This can be achieved by examining the interface (Liu et al.,
2022) between the SPG interior where much of the buoyancy forcing takes place and the narrow, swift boundary currents that rapidly transmit this information around the SPG and enable connections with other basins (Fig. 1). Along-slope density gradients result in geostrophic flow towards the boundary from the interior, driving an intensified along-slope flow (Huthnance et al., 2020; Marsh, 2017). The boundary currents in turn determine the basin-wide density gradients through the thermal-wind relation and these set the structure, strength and
variability of the transocean AMOC, driving meridional fluxes of heat, freshwater and carbon.

Over several decades there has been a focus on the interior-boundary current exchanges using idealised models investigating the eddying and isopycnal pathways into and out of the boundary, surface buoyancy forcing and wind forcing. The boundary current can contribute directly to poleward heat transport through along-isopycnal transport (Straneo, 2006). The AMOC can also be influenced by interior-boundary exchange processes (Spall,
2008), changes in water mass properties in the boundary currents (Williams et al., 2015), and interaction between boundary currents and steep topography driving diapycnal mixing (Brüggemann and Katsman 2019; Liu et al., 2022; Spall and Pickart, 2000).

Recent model and regionally based studies also focus on the importance of extreme buoyancy forcing in the interior and exchanges with the boundary currents. Deep winter mixing has the capacity to influence the
AMOC and is thought to be dependent on atmospheric forcing (De Jong et al., 2018; Josey et al., 2019), and on diapycnal mixing between the interior and the boundary via eddy diffusion. Water transformed to higher densities in the basin interior requires a mechanism for export into the boundary current, and eddies may provide a means for this to happen in upper and intermediate waters (Le Bras et al., 2020). However, as deep mixing relies on weak stratification the intrusion of warm, salty intermediate water masses (De Jong et al.,
2018) or freshwater input from the boundary current can act to inhibit convection. Le Bras et al., (2021) and Zhang and Thomas, (2021) show that freshwater from Fram Strait has a more active role in mediating the AMOC than Greenland runoff, which may remain in coastal waters in the vicinity of Greenland. McDougall and Ferrari (2017) also show the globally important role of thin boundary layers to drive the downward heat flux necessary for closing overturning circulations.

To synthesise regional studies of interior-boundary interaction we calculate a budget for the exchange of water between the SPG interior and boundary/shelf regions, and through a zonal transatlantic section at 47° N (Fig. 1). We construct a new temperature-salinity (TS) climatology along the 1000 m depth contour of the SPG and closing at 47° N (12,000 km path, Fig. 1) covering the Argo era (2000 onwards).

The 1000 m isobath was selected for numerous reasons. Firstly, the 1000 m contour encircles the key features of
the subpolar gyre, including the Rockall, Iceland, Irminger and Labrador basins, partitioning basin interior processes from shelf sea processes. Secondly, at 47 °N the simulated maximum overturning in depth space is roughly 1000 m depth (Hirschi et al., 2020), so this choice allows us to approximately distinguish upper and lower limb processes. Thirdly, Argo trajectories allows us to estimate currents at 1000 m depth which we later incorporate into our analysis.





We quantify regionally and in density space where the volume transports into and out of the SPG interior occur.
We then validate and extend our analysis using the VIKING20X model (Biastoch et al., 2021; Fox et al., 2022),
which, when combined with our new climatology provides novel insights into the functioning of the AMOC in
the SPG. Finally, we present the overturning, heat and freshwater fluxes associated with the observed water
properties and transports.

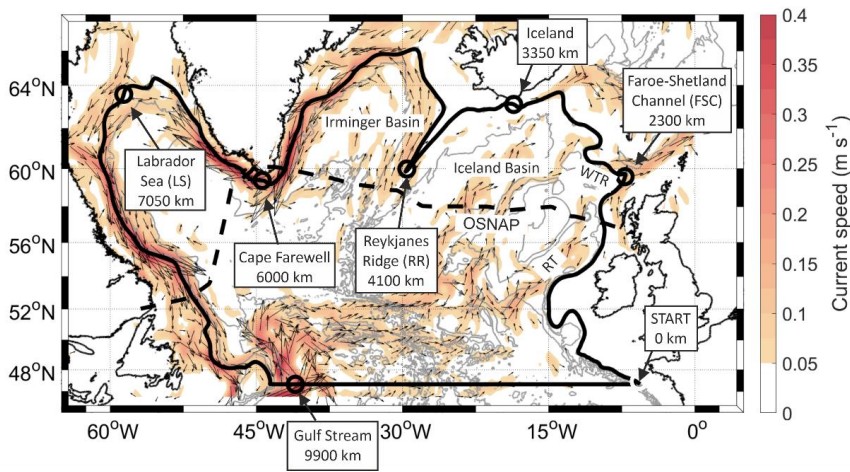


*Figure 1. Smoothed 1000 m bathymetry contour (solid black line), closed by transect across 47° N. Key
locations around contour are labelled; these are used throughout this study. Dashed black line shows OSNAP
line. RT: Rockall Trough, WTR: Wyville Thomson Ridge. Mean magnitude and direction of surface currents
(2000-2020) derived from AVISO data shown by coloured contours and quiver arrows. Isobaths overlaid at*
*1000 m increments. Bathymetry contours from GEBCO bathymetry (http://www.gebco.net/). GEBCO =
General Bathymetry Chart of the Oceans.*

**2. Materials and methods**

Here we describe the datasets and methods used for the core analyses in the study. Information on other datasets
used is provided in Supplementary Materials S2.

**2.1. World Ocean Database (WOD18) profile data**

We construct our TS climatology along a narrow strip defined by the 1000 m isobath around the basin of the
SPG. CTD and Argo profile data from post-2000 (Argo era) were downloaded from the WOD on 03/09/2019
(Boyer et al., 2018). The isobath was smoothed using a 100 km along-contour bracket to remove undesired
complexity in the contour and profiles of conservative temperature ($\theta$) and absolute salinity ($S$) were gathered
between 0 and 75 km offshore as shown in Fig. 2. We required data coverage between surface and 1000 m so
profiles with poor vertical resolution (< 50 observations), and those sampling only part of the water column,
were excluded. Further QC steps were performed and are detailed in Supplementary Materials S1.



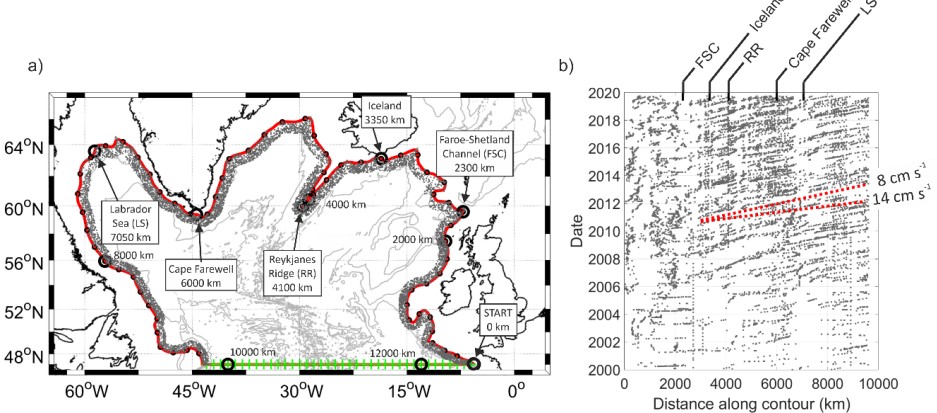

*Figure 2: (a) Location of profiles contributing to basin perimeter data product. 1000 m bathymetry contour is*

*shown in red. Profiles extend up to 75 km offshore from the 1000 m contour. Black points on boundary contour*

*show grid locations of boundary data product, green crosses show grid locations of EN4 zonal section at 47° N.*

*Bathymetry contours from GEBCO bathymetry (http://www.gebco.net/). GEBCO = General Bathymetry Chart*

*of the Oceans. (b) Distribution of profiles contributing to boundary dataset plotted over time. Dashed red lines*

*show nominal speeds of 8 cm s$^{-1}$ and 14 cm s$^{-1}$ around the boundary. Key locations around boundary labelled.*

*FSC: Faroe-Shetland Channel, southernmost point of Iceland, RR: Reykjanes Ridge (southern tip), Cape*

*Farewell, LS: Labrador Sea.*

### 2.2. Gridding of profile data

Profiles were first separated into four seasons: Winter (JFM); Spring (AMJ); Summer (JAS); and Autumn

(OND). They were gridded vertically in 20 dbar pressure bins and then horizontally. For the horizontal gridding

we used cells spaced at regular 150 km intervals and employed a variable search radius centred on each cell. For

a given grid cell, an initial search radius of 150 km was used, and the number of profiles found in this radius of a

cell evaluated. If 75 raw profiles were not found, this search radius was incrementally expanded up to a

maximum of 300 km. Thus, some profiles are used in more than one grid cell. Most grid cells are populated

using the minimum search radius (± 150 km), but it was necessary to expand the search radius up to the

maximum 300 km to achieve good coverage in 5 % of cells during the summer, rising to 22 % during the

autumn. Profiles were averaged on pressure levels to create the gridded product of Θ and S. A schematic of the

gridding workflow is provided in Supplementary Materials S1.

### 2.3. EN4 data at 47° N

We use temperature and salinity data from the Met Office EN4 product (Good et al., 2013) for the zonal section

to close the boundary at a latitude of 47° N. We considered this to be the most appropriate source of data for the

zonal transect: first, whilst our boundary dataset benefitted from an 'along-boundary' gridding methodology, the

zonal transect is aligned to EN4's grid, so the benefits of independently gridding the profile data are largely

negated. Second, EN4's climatology provides coverage deeper than 2000 m in the North Atlantic, a region

where observational data is sparse due to the depth limit of most Argo floats.



We found excellent agreement between gridded profiles and EN4 grid cells in <2000 m waters, and no unusual horizontal gradient in properties (which could translate into an anomalous geostrophic transport) between the end of the boundary dataset and the beginning of the EN4 transect. The location of WOD profile data and EN4 grid cells is shown in Fig. 2.

### 2.4. Computing transports and fluxes

**2.4.1. Geostrophic velocities**

We first compute the geostrophic shear between each gridded station, and between the final station and the first to complete the loop. Note that when integrating to the same depth around the loop, the net transport between the interior and exterior of the SPG is constrained to be near-zero because there is no net change in dynamic height around the closed circuit. A small residual transport remains because of variations in the Coriolis

parameter $f$ as the latitude of the stations changes around the boundary (the 'beta effect').

When computing overturning transport and heat and freshwater fluxes in Sect. 3.4 and 3.5, we require a measure of transports to the seabed so that volume is conserved on completion of the boundary loop. Geostrophic velocities across the >1000 m depth of the 47° N transect result in a net gain in volume by the SPG interior, so we enforce the conservation of volume using a small negative reference velocity applied to this region. The

EN4 dataset is known to poorly resolve the deep western boundary current in this region (Fraser and Cunningham, 2021) which probably explains the imbalance. The implementation of this reference velocity, and its impact on computed values for fluxes and overturning is discussed in Sect. 2.4.5.

Dynamic height at each profile is computed relative to the surface and referenced to the gridded Absolute Dynamic Topography (ADT) derived from satellite altimetry (Eq. (1)). We consider the use of satellite SSH-

derived velocities to be a robust reference method for our application given the large spatial scales and the long temporal averages associated with the study. The gridded ADT data were temporally averaged over the same periods as the profile data coverage (2000-2019, split into four seasons) and interpolated values extracted at the station locations. They were then smoothed using a 5-point running average to mimic the smoothing inherent in the hydrographic gridding process.

$$\Phi_{total} = \Phi_{bc} + g\eta \ [m^{-2} \ s^{-2}]$$
(1)

Where $\Phi_{bc}$ is the dynamic height relative to the sea surface, calculated as the integral of the specific volume anomaly from the gridded pressure to the surface. $\eta$ is the satellite-derived ADT and $g$ is acceleration due to gravity. The time-mean geostrophic velocity assigned to locations mid-way between hydrography stations between $v_{geo}$ is computed from:

$$v_{geo} = \frac{1}{f} \frac{d\Phi_{total}}{dx} \ [m \ s^{-1}]$$
(2)

where $x$ is the along-contour coordinate.





### 2.4.2. Geostrophic transports

Transports for each grid cell were computed by integrating Eq. (2) over the cross-sectional area between each
station, and between adjacent pressure levels (the 20 dbar pressure intervals are taken to approximate 20 m):

$$Q_{grid\ cell} = \iint\limits_{cell\ area} v_{geo}\ dxdz\ [Sv] \tag{3}$$

The vertically integrated transport between 0 and 1000 m can then be computed by summing the transports of
cells at each station. Further, the cumulative transport around the basin can be obtained using a horizontal
cumulative integral.

We estimate uncertainties based on the variability inherent in the datasets contributing to the study. This is
accomplished by repeating the analysis multiple times with the gridded TS profiles randomly perturbed. The
perturbation of each gridded value is scaled by the standard deviation of profile data contributing to that grid
cell, thus giving an indication of the sensitivity of the conclusions to the scatter of 'raw' profiles. For the EN4
transect, the uncertainty is supplied with the gridded variables, and we use this to scale the perturbations. The
satellite altimetry has a large standard deviation on day or month timescales. As our analysis spans two decades,
we considered it appropriate to first calculate annual means of ADT, then compute the standard deviation of
these annual means for the uncertainty estimate. The analysis was repeated 100 times with the boundary
climatology, altimetry, and surface Ekman transports (Sect. 2.4.3). The standard deviation of the resultant values
forms the upper and lower bounds supplied with our results.

### 2.4.3. Surface Ekman transports

Wind stress data was obtained from the ECMWF ERA5 reanalysis product (Hersbach et al., 2020). The wind
stress component tangent to the boundary contour was used to calculate the Ekman transport across the
boundary at geostrophic velocity locations (cell mid-points). These were then averaged to compose seasonal
climatologies. Uncertainties associated with the surface Ekman transports were taken as the standard deviation
of annual means of the transports.

### 215 2.4.4. Model-derived transports in VIKING20X

We recreate the boundary transect in VIKING20X to support the observational analysis and help diagnose the
transports and fluxes which may not be resolved by geostrophic or surface Ekman calculations. Output of the
VIKING20X-JRA55-short model hindcast (Biastoch et al., 2021) is used to compute transports into the SPG.
VIKING20X is a 0.05° ice/ocean model of the Atlantic Ocean (33.5° S to ~65° N) nested within a 0.25 degree
global ice/ocean model. The run used here is driven from 1980-present using JRA55-do atmospheric forcing and
runoff (Tsujino et al., 2018). In the vertical, VIKING20X uses 46 geopotential z-levels with layer thicknesses
from 6 m at the surface gradually increasing to ~250 m in the deepest layers. Bottom topography is represented
by partially filled cells allowing for an improve representation of the bathymetry (Barnier et al., 2006). In the
SPNA VIKING20X has horizontal resolution of 3-4 km. Hindcasts of the past 50-60 years in this eddy-rich
configuration show it realistically simulates the large-scale horizontal circulation, the distribution of the





mesoscale, overflow and convective processes, and the representation of regional current systems in the North and South Atlantic (see Biastoch et al., 2021 for full details).

To preserve the volume conservation in VIKING20X, rather than mimicking the observational data sampling the transport calculations are performed across a section following horizontal grid-cell boundaries (T-grid

boundaries in the VIKING20X ocean Arakawa C-grid). North of 47° N this section is constructed to be the shallowest line with all adjacent cells deeper than 1000 m, the volume is closed across 47° N. Total model transports, model geostrophic transports (referenced to model sea surface height), and model surface Ekman transports are calculated.

The stepped model topography results in two potential approaches for estimating geostrophic transports. The

first stops strictly at 1000 m but leaves a small gap beneath over complex bathymetry. This approach obeys the beta constraint on geostrophic flow, so is most comparable to the observations but some 'leakage' below 1000 m on the boundary remains. The other approach extends to the bed around the boundary. This means that all across-boundary flow is captured, but the beta constraint on total geostrophic transport is slightly relaxed as there is now an undulating bed with along-section pressure differences. When comparing observations to

VIKING20X (Sect. 3.3) we primarily use transports derived using the strict 1000 m cut-off. However, when estimating the gyre volume budget (Sect. 4.4) we compute transports to the seabed around the boundary as this enforces a strict separation of flows across the 47° N transect.

To diagnose ageostrophic near-bed flow associated with the modelled boundary current, an estimate of model bottom Ekman transport $Q_{EB}$ (per unit section length) into the SPG is made:


$$Q_{EB} = \frac{C_d \cdot \sqrt{u^2 + v^2 + e_b} \cdot u}{f} \tag{4}$$

from model parameters $C_d$=0.001, $e_b$=0.0025 m$^2$ s$^{-2}$. $C_d$ is the bottom drag coefficient, and $e_b$ the bottom turbulent kinetic energy loss due to tides, internal waves breaking and other short time scale currents. $u$ is along-section velocity, $v$ is velocity perpendicular to the section and $f$ is the Coriolis parameter.

### 2.4.5. Heat and freshwater fluxes

For this analysis the gridded temperature and salinity are interpolated onto the 'mid-point' geostrophic velocity stations and σ$_0$ recalculated. The computation of fluxes requires a mass-balanced velocity field, and this necessitates computing transports down to the seabed rather than for the top 1000 m only. Whilst we have confidence that geostrophic + surface Ekman capture the main flow features of the upper ocean, as previously stated we consider that geostrophic shear using the EN4 TS fields does not adequately resolve several features

of the deep flow across 47 N. Computing cumulative geostrophic + surface Ekman transports for the full depth results in residuals averaging +20 Sv into the SPG, mainly because the Gulf Stream does not diminish with depth, but also due to an underestimation of the deep western boundary current, and an absence of southern flow in the deep water masses across 47° N. We therefore perform a 2-stage adjustment to the sub-1000 m velocities to first linearly reduce the Gulf Stream with depth, then add a seasonally varying reference velocity that when

added to the 47° N section (integrated between 1000 m and seabed) balances the water volume entering and



leaving the SPG. This is between -0.0002 and -0.0018 cm s$^{-1}$ depending on season. Details of this adjustment are provided in Supplementary Materials S5.

Heat and freshwater fluxes across the boundary were calculated as follows. Heat flux ($Q_\theta$) across each grid cell is defined as:


$$Q_{\theta_{grid\,cell}} = \rho C_p \iint_{cell\,area} v_{total}\,(\theta - \bar{\theta})\,dxdz\,[W]$$

(5)

Where $\rho$ is the nominal potential density of seawater, $C_p$ is the specific heat of seawater, $v_{total}(x, z)$ is the sum of the geostrophic (Eq. (1)) and Ekman velocities (Sect. 3.2.2) perpendicular to the section, $\theta(x, z)$ is the conservative temperature and $\bar{\theta}$ , the reference temperature, is the mean temperature for the full-depth SPG

interior (4.03 °C). Following Lozier et al., (2019) we use a value of 4.1 x 10$^6$ Jm$^{-3}$ K$^{-1}$ for $\rho C_p$.

Freshwater flux ($Q_f$) is defined as:

$$Q_{f_{grid\,cell}} = - \iint_{cell\,area} v_{total}\,\frac{S - \bar{S}}{\bar{S}}\,dxdz\,[m^3 s^{-1}]$$

(6)

Where $S(x, z)$ is the absolute salinity of a grid cell, $\bar{S}$, the reference salinity, is the mean salinity for the full-depth SPG interior (35.14 g kg$^{-3}$). As before the convention for $Q_\theta$ and $Q_f$ is positive into the SPG.

We estimate the average surface freshwater and heat fluxes for 2000-2019 using ERA5 month means (Hersbach et al., 2020). For freshwater we compute evaporation – precipitation for each grid cell, then integrate over the total surface area enclosed by the 1000 m contour and 47° N (4.6x10$^6$ km$^2$) using an area-weighted mean. We calculate downward surface heat flux as the sum of sensible, latent, shortwave, and longwave heat fluxes. Surface flux errors are estimated as the standard error of the annually averaged timeseries for the summed

components following Li et al., (2021a).

### 2.4.6. Eddy Kinetic Energy and boundary topography

Eddy kinetic energy (EKE) was calculated from satellite ADT for the period of study using

$$EKE = \overline{{u'_s}^2 + {v'_s}^2}$$

(7)

where $u'_s$ and $v'_s$ are the high-frequency components (150-day highpass filtered) of the surface geostrophic

velocity components along the SPG boundary contour. The overbar denotes seasonal averaging to form climatologies.

Seabed slope angle was calculated from 30 arc-second GEBCO bathymetry on the native grid (GEBCO compilation group 2019) then interpolated onto ~1 km horizontal resolution rendition of the 1000 m depth contour (derived from the same GEBCO data set). A 480-point moving mean was applied along contour. Slope

is a scale dependent quantity: at the visual map scale a 480-point running mean does not equate to a 480km straight line moving average since at 1 km scale the 1000 m contour is highly irregular.



## 3. Results

### 3.1. Hydrography

The counter-clockwise evolution of water properties around the closed SPG boundary is shown in Fig. 3, and a
full-depth section across 47 ̊N is shown in Fig. 4. These figures depict the annual average water properties;
seasonal anomalies are supplied in Supplementary Materials S3.

In general, the density at a given depth level increases with progress along the 1000 m isobath. By the thermal
wind relation, the geostrophic shear is therefore typically negative (i.e. driving export across the boundary out
from the interior). Between the boundary start near Biscay and the Faroe-Shetland Channel (FSC) the water
column is thermally stratified and this controls the density distribution (salinity changes only gradually with
depth). Between 1000 and 2000 km (European Shelf), the along section density gradient at a fixed depth is
positive shallower than 750 dbar and is negative deeper than 750 dbar. This is consistent with the expected
density evolution of the adjacent slope current in this region (Huthnance et al., 2022). The horizontal density
gradient increases at the entrance to the FSC. Between here and Iceland, a persistent negative geostrophic flow,
strongest near the surface, is associated with a thermally driven positive density gradient. Between Iceland and
Cape Farewell, further cooling, freshening and densification occurs throughout the water column. Geostrophic
flow is largely out of the SPG shallower than 500 dbar, and into the interior below 500 dbar.

Cape Farewell marks the beginning of a pronounced change to the water column structure. A very cold and
fresh surface layer results in a low density layer shallower than 250 dbar. There is also a negative horizontal
density gradient below 250 dbar, but this is driven by an increase in temperature with progress around the gyre.
West of Cape Farewell (i.e. along West Greenland) there are positive geostrophic flows which can be attributed
to a portion of the West Greenland Current (WGC) crossing into the SPG, switching to negative in the north-
eastern Labrador Sea. In the north-western Labrador Sea, the trend towards increasing density is resumed, this
time driven by further cooling below 250 dbar.

Geostrophic flow in the north-western Labrador Sea is into the SPG and is greatest at depth. The influence of
the cold, Labrador Current in the surface layers extends along the Newfoundland and Labrador shelf edge as far
as 47° N. Horizontal density gradients are very weak over this region, consequently geostrophic flow is near-
homogeneous with depth. The boundary tracks the northern rim of the Flemish Cap before crossing the North
Atlantic at 47° N. The Labrador current is bisected here as it exits the SPG. The Gulf Stream is clearly visible
on the western side of the 47° N transect as a narrow region featuring rapid warming and salinification driving a
steep negative horizontal density gradient. This is associated with a region of very strong barotropic flow into
the interior and strong flow out of the interior immediately adjacent. Thermal wind results in a reduction of
current strength with depth. East of the Gulf Stream system, the zonal transect is largely characterised by
positive geostrophic flow northward and weakening with depth.



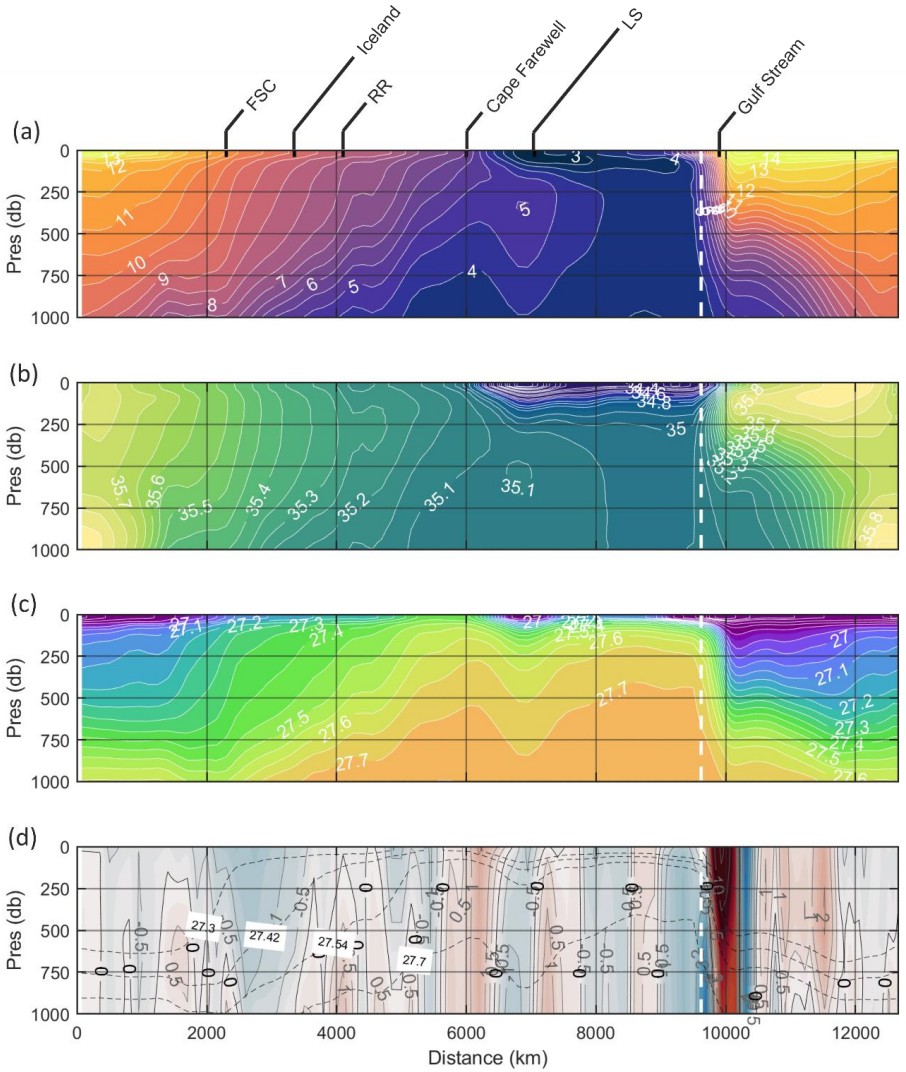


*Figure 3: Gridded boundary product plotted by distance along the 1000 m contour travelling anti-clockwise around the basin. Annual means shown. (a) conservative temperature (°C), (b) absolute salinity (g kg⁻¹), (c) density (σ₀, kgm⁻³), (d) geostrophic velocities across the boundary perpendicular to the 1000 m depth contour (cm s⁻¹, positive into the interior, negative out of the interior, colour map intervals of 0.25 cm s⁻¹ with selected*

*contours shown). Density contours relevant to overturning processes (Fig. 9) shown by black dashed lines. The transition to the 47° N section, and from gridded CTD to EN4 climatology data, is delineated by the dashed white line. Key locations around boundary labelled. FSC: Faroe-Shetland Channel, southernmost point of Iceland, RR: Reykjanes Ridge (southern tip), Cape Farewell, LS: Labrador Sea, Gulf Stream.*





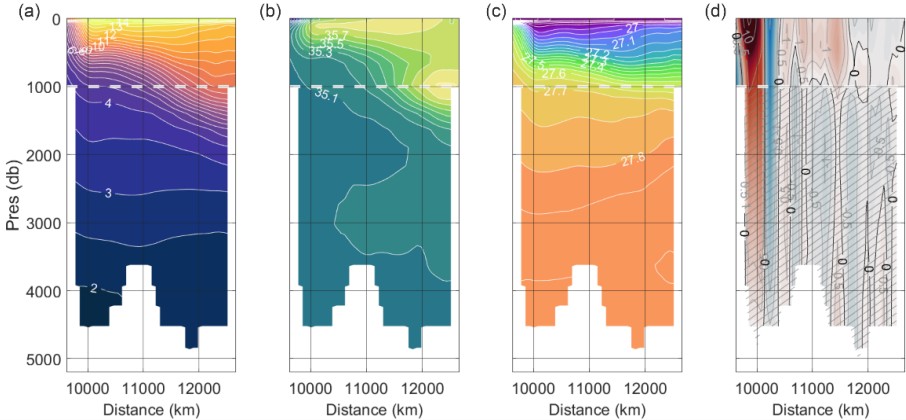

*Figure 4: Zonal (47° N) section constructed from EN4 data and shown to full depth for (a) conservative
temperature (℃), (b) absolute salinity (g kg⁻¹), (c) density (σ0, kgm⁻³), (d) geostrophic velocities into the SPG,
perpendicular to contour (cm s⁻¹). Annual means shown. The sub-1000 m region in (d) delineated by cross-
hatching is subject to a correction velocity (see Supplementary Materials S5 for details). The 1000 m vertical
threshold for transport calculations is delineated by dashed white line.*

**3.2. Transports perpendicular to boundary**

**3.2.1. Geostrophic transports above 1000 m**

The depth integrated geostrophic transport across the boundary (Fig. 5a) is broadly out of the SPG in the FSC
and to the east of Iceland, and to the west of the Gulf Stream. Inflow is dominated by northward flow across 47°
N (above 1000 m) mostly in the Gulf Stream (+20 to 30 Sv) but also across the width of the Atlantic. However,
within this there are striking regional patterns of inflow and outflow, and regions where there is only limited
flow across the boundary. Along the European continent there is outflow south of Ireland and then inflow to the
north, perhaps a suggestion of cyclonic circulation over Goban Spur at shallower depths. North of Ireland some
outflow is evident, suggesting transport onto the Malin and Hebridean shelf (Jones et al., 2018; Jones et al.,
2020; Porter et al., 2018). Between Scotland and Iceland, -10 to -12 Sv of outflow marks the exit of the NAC
over the Wyville Thompson Ridge. Around the Reykjanes Ridge the pattern of flow is consistent with net
westward cross-ridge flow quantified by Petit et al., (2018). Flow in the vicinity of Cape Farewell is notable for
the large transports into the SPG associated with the East Greenland Current (EGC) and its retroflection (e.g.
Holliday et al., 2007) while outflow in the north-eastern Labrador Sea is the result of a portion of the WGC
exiting the SPG towards Davis Strait. Approximately the same volume re-enters the SPG along north Labrador
as a portion of the Labrador Current which flows parallel to the boundary down the Labrador and Newfoundland
shelf and shelf edge (Lavender et al., 2000). From about 7800km to west of the Gulf Stream sustained outflow
results in a net export of -12 Sv onto the Labrador Shelf.

Along 47° N east of the Gulf Stream inflow there is a narrower region of recirculating outflow, then a weak
inflow across most of the section to the east. The net cumulative transports into and out of the SPG return to



near-zero on completion of the circuit, with a small positive residual in all seasons (+2.13 to +2.58 Sv) due to the beta effect. Cumulative geostrophic transports above 1000 m are shown in Table 1.

Seasonal transport variations are relatively small (Fig. 5b). Between the FSC and the western Irminger Sea (5500 km) autumn and winter transports out of the SPG are 1-2 Sv greater than spring and summer before converging at Cape Farewell. Similarly, along the Labrador Seaboard autumn and winter transports out of the SPG tend to be greater than those in spring and summer but converge when crossing the Gulf Stream.

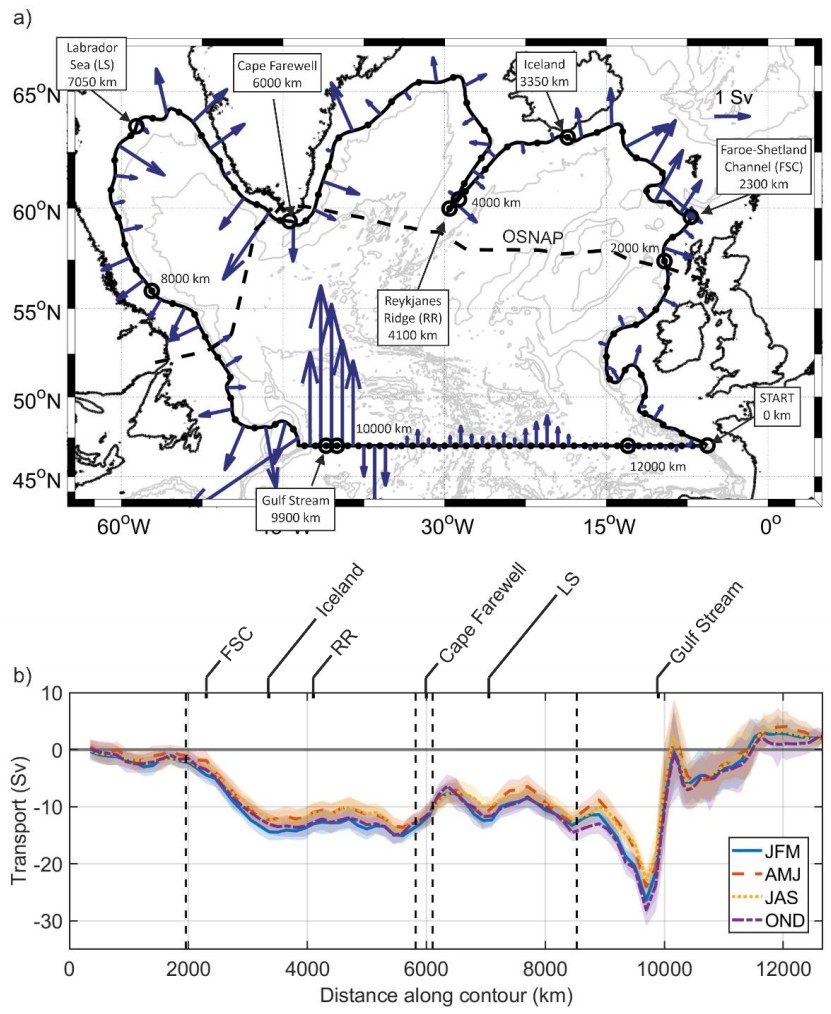

*Figure 5: Geostrophic Transport perpendicular to contour, into the SPG positive. a) Depth integrated volume transport between 0 and 1000 m for each grid cell (timeseries/annual mean). Horizontal bins are 150 km apart around the 1000 m contour, and 1° across 47° N. Bathymetry contours from GEBCO bathymetry (http://www.gebco.net/). GEBCO = General Bathymetry Chart of the Oceans. (b) Cumulative volume transport around basin, for each season. Key locations around boundary labelled as for Fig. 3, vertical dashed lines denote OSNAP crossings.*



### 3.2.2. Surface Ekman transport perpendicular to boundary

Due to the prevalent cyclonic weather systems over the SPNA surface Ekman transport is generally directed out

of the SPG, with winter exhibiting the largest transports and summer the weakest (Fig. 6). South-westerly winds in the north-eastern Atlantic result in net transports out of the SPG onto the continental shelf west of the British Isles. Between Scotland and Iceland there is little surface Ekman transport across the boundary, due mainly to the prevailing surface Ekman transport being roughly parallel to the boundary contour rather than lower wind speed (e.g. Laurila et al., 2021). Conversely, very high transports out of the SPG off south-east Greenland are

due both to energetic storm systems and to the boundary contour being approximately perpendicular to prevailing surface Ekman flow. There is strong seasonality off south-east Greenland, with cumulative transports varying from -0.5 in the summer to -2 Sv in the winter. Off south-west Greenland there is net inflow into the SPG, except in summer. This is the only location that sees seasonal sign reversal (+1 Sv in winter to -0.2 Sv in summer). While the Labrador Sea gains volume off south-west Greenland during the winter, between Cape

Farewell and the OSNAP crossing at 8500 km there is a net loss of -1.8 Sv in the winter compared to -0.1 Sv in the summer, resulting in a large seasonal signal from this region. Between the western Labrador Sea and the Gulf Stream, surface Ekman transports are almost exclusively out of the SPG. This trend continues across the 47° N section, with a further strengthening of the net seasonal signal due to weak spring and summer negative transports contrasting with strong autumn and winter transports. Net surface Ekman transports out of the SPG

range from -2.45 Sv in the summer to -7.70 Sv in the winter (Table 1).



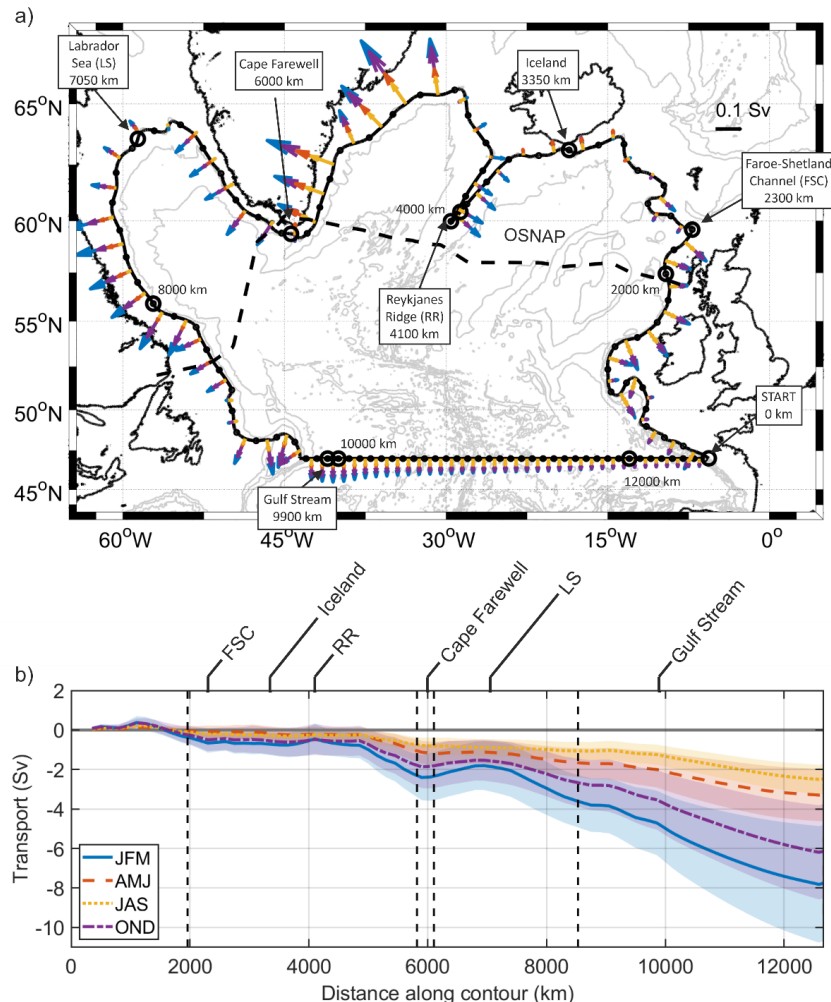

*Figure 6: Surface Ekman Transport perpendicular to contour, into the SPG positive. (a) volume transport between 0 and 1000 m for each grid cell, coloured by season. Horizontal bins are 150 km apart around the 1000 m contour, and 1° across 47° N. Bathymetry contours from GEBCO bathymetry (http://www.gebco.net/). GEBCO = General Bathymetry Chart of the Oceans. (b) Cumulative volume transport around basin, for each season Key locations around boundary labelled as for Fig. 3, vertical dashed lines denote OSNAP crossings.*

The surface Ekman transports, when summed with the geostrophic transports contribute a marked seasonal component to the net cumulative transport across the boundary. In winter there is a -5.36 Sv residual transport out of the SPG, whereas in the summer the residual reduces to -0.32 Sv (Table 1), with the seasonal range driven almost entirely by the surface Ekman component.




*Table 1: Net transports into SPG above 1000 m, in Sv.*

|  | Geostrophic [Sv] | Surface Ekman [Sv] | Geostrophic + Surf. Ekman [Sv] |
|---|---|---|---|
| **Annual mean** | 2.29 ± 0.09 | -4.87 ± 1.82 | -2.57 ± 1.92 |
| **JFM** | 2.33 ± 0.09 | -7.70 ± 2.90 | -5.36 ± 2.99 |
| **AMJ** | 2.15 ± 0.09 | -3.25 ± 1.32 | -1.10 ± 1.41 |
| **JAS** | 2.13 ± 0.09 | -2.45 ± 0.73 | -0.32 ± 0.82 |
| **OND** | 2.58 ± 0.10 | -6.09 ± 2.36 | -3.51 ± 2.46 |

### 3.2.3. Bottom Ekman transport

Bottom Ekman transport is an essential dynamical feature of cyclonic ocean boundary (slope) currents

(Huthnance et al., 2020), and a significant transport mechanism from slope regions to adjacent ocean interior (Huthnance et al., 2022). Typical slope boundary current velocities range from a few to several 10s of cm s$^{-1}$. We make an approximate observation-based estimate of the bottom Ekman transport by examining the tracks of Argo floats contributing to the boundary dataset (Fig. 2b). Diagonal stripes, particularly after 3000 km indicates advection of the floats around the SPG boundary at the typical 1000 m drift depth at 8 to 14 cm s$^{-1}$. We

therefore take as a lower bound an along-slope velocity of 8 cm s$^{-1}$ around the SPG boundary to approximate the bottom Ekman transport $Q_{EB}$ into the SPG following theoretical arguments by Souza et al., (2001) and Simpson and McCandliss, (2013):

$$Q_{EB} = \frac{k_b \hat{v}^2}{f} \ [m^{-2}s^{-1}] \qquad (8)$$

where $k_b$ is a bottom friction coefficient (taken as 0.0025 following Simpson and McCandliss, 2013), $\hat{v}$ is the

mean along-slope velocity and $f$ is the local Coriolis parameter. As $f$ varies around the boundary we compute $Q_{EB}$ for each grid cell on the boundary and integrate horizontally. This results in a total transport into the SPG of 1.3 Sv for the lower bound. This estimate increases to 3.8 Sv when using a figure more representative of an upper bound on an average along-slope velocity of 14 cm s$^{-1}$.

Given the large uncertainties associated with the observation-based bottom Ekman estimates we exclude this

process in the transports contributing to the overturning and flux totals, however it is relevant to the discussion of the SPG volumetric budget. The potential contribution of bottom Ekman transport and other near-bed processes to the SPG volume budget is discussed in Sect. 4.4.

### 3.3. Transports perpendicular to boundary in VIKING20X

The 20-year mean geostrophic volume transports into the SPG calculated from the VIKING20X model

hydrography show broad agreement with the observation-based geostrophic transports at large spatial scales (Fig. 7a). Both show outward transport of 25-30 Sv round the 1000 m contour, balanced by inward transport in the surface 1000 m across 47° N, with a total net inflow of 2-3 Sv around the full perimeter. The geostrophic transport of the Gulf Stream above 1000 m is about 25 Sv in both the model and observations. Between the FSC





and Iceland, 12 Sv is exported out of the SPG in the observed transports, and 9 Sv is exported in the model. The
modelled and observed transports then converge and exhibit little difference between the RR and Cape Farewell.

At smaller spatial scales there are dissimilarities between model and observational geostrophic flows.
VIKING20X has steady outflow through FSC, and east & west of Iceland, contrasting with the observations
which show outflow focussed in the FSC and Iceland Faroe Ridge (2000-3000 km), though the total transport
accumulated between the FSC and the OSNAP crossing east of Cape Farewell is the same in each case. There is
a contrast in behaviour around Labrador Sea. VIKING20X shows 5 Sv spatially uniform outflow from the
interior between Cape Farewell and the western end of OSNAP, whilst the observations show alternating
regions of inflow and outflow round the Labrador Sea. In particular, the model has no inflow to match
observations at the northern half of west Greenland. There are also different patterns of inflow across 47° N,
with the model showing stronger inflow in the region east of the main Gulf Stream core but west of the Mid-
Atlantic Ridge (10300-11000 km).

There are many possible reasons for these differences and a detailed examination is beyond the scope of the
current work. The Labrador Sea boundary is a region of very steep topography, complex interactions, and
poorly understood freshwater influence. Model TS and density structure in this region may be unrealistic (see
Biastoch et al., 2021). The model section and the observational climatology are constructed quite differently
along the 1000 m section: the model hydrography is sampled along a single line closely following the 1000 m
contour, while the observations involve spatial averages over large (75 km x 150 km) areas offshore of this
contour. Observed geostrophic velocities are referenced to satellite-derived ADT at the surface, these are at
coarser resolution than the dynamically consistent modelled sea surface heights used to reference the
VIKING20X geostrophic velocity calculation.

In calculating transports from the observation-based climatology we consider the contribution from geostrophy
in the top 1000 m and surface wind forcing. The SPG boundary features regions of steep topography, strong
boundary currents, deep overflows and enhanced eddy activity and it is therefore unclear how much of the total
cross-boundary flow we are capturing. We can use the model results to look at details of the missing transports.
Figure 7a shows two candidate processes: bottom Ekman layer frictional flows, and the remainder primarily
driven by nonlinear and viscous processes. In Fig. 7b we examine the possible missing transports due to flows
beneath the base of our 1000 m contour and the bed (due to the observed climatology being on average offset
from the continental slope, see Fig. 2). In VIKING20X this is achieved by integrating to the bed along the 1000
m contour, as opposed to using a strict 1000 m cutoff. The difference arises due to the stepped model
topography and associated inability to follow the 1000m bathymetry precisely.

The results show that over most of boundary the across-boundary flows are dominated by geostrophic flows
(orange/dashed orange lines, Fig. 7a and b). The major exceptions are the deep overflow regions of the Denmark
Strait and east Greenland (around 5000-5500 km) and, to a smaller extent, the Faroe Bank Channel overflow at
around 3000 km. While these unobserved processes dominate the cross-boundary transports at two locations
they account for the majority of the cross-boundary transport when integrated round the whole boundary above
1000 m (grey/dashed grey lines, Fig. 7a and b). We discuss this further in Sect. 4.5.



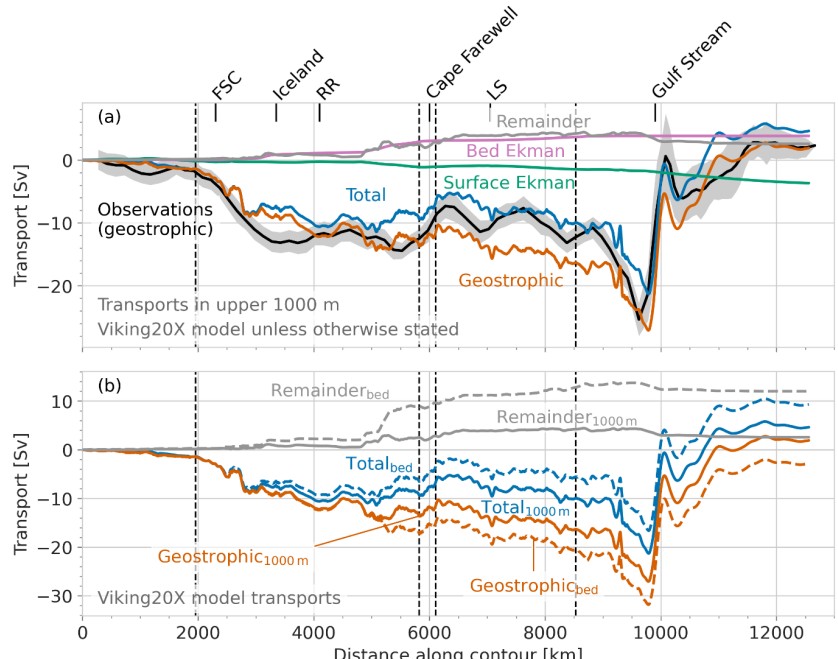

*Figure 7: (a) Comparison between observed cumulative geostrophic volume transport into the SPG (annual mean) and transport components above 1000 m in VIKING20X. Key locations around boundary labelled as for Fig. 3, vertical dashed lines denote OSNAP crossings. Remainder is calculated as (total – geostrophic -surface*

*Ekman – bed Ekman). (b) Comparison between transports into the SPG in VIKING20X using a strict 1000 m cutoff, and integrating to the bed along the 1000 m contour (but still integrated to 1000 m across 47° N). The difference is due to the stepped model topography and associated inability to follow the 1000m bathymetry precisely.*

### 3.4. Overturning in the Subpolar Gyre

Here we compute the density-space overturning circulation in the SPG using the sum of the observed geostrophic and surface Ekman fluxes. For this analysis it is necessary to integrate to the seabed across the 47° N transect so we apply a reference velocity below 1000 m on the 47° N transect to enforce the conservation of volume (Supplementary Materials S5). The full-depth transports are shown in Fig. 8. As the adjustment is applied to waters below 1000 m, it almost exclusively impacts lower limb flows, and is below the main features

of the overturning stream function (Fig. 9).

The full-depth transports in Fig. 8 are divided into upper, intermediate and lower layers based on density thresholds established using inflection points in the overturning stream function (Fig. 9). These density thresholds are also overlaid on the geostrophic velocities depicted in Fig. 3d. The upper layer has a net gain of 7.36 Sv, while the intermediate and lower layers have a net loss of -3.85 Sv and -3.53 Sv respectively. The

upper layer loses volume around the SPG from Biscay to the Gulf Stream. This loss of ~2.5 Sv occurs around 2000 km and is associated with surface cooling and exchange with the European shelf. Approximately 0.5 Sv

(20 %) of the loss is due to air-sea heat exchange, and the remaining 2 Sv (80 %) is advected out of the SPG by geostrophic and Ekman flows.

West of the FSC, the water column is sufficiently dense that the upper layer makes little further contribution to total transport. The intermediate layer accounts for almost all the NAC transport out of the interior across the Wyville Thompson Ridge. As the boundary contour advances around the Irminger Basin (4100-6000 km), the lower layer becomes the dominant contributor to total transport. The lower layer accounts for most of the inflow in the vicinity of Cape Farewell and the subsequent regions of inflow and outflow in the Labrador Sea and along north Labrador. The export of water west of the Gulf Stream is also almost entirely in the lower

layer. The Gulf Stream drives transport into the interior in all layers, though the contribution is largest in the lower layer.

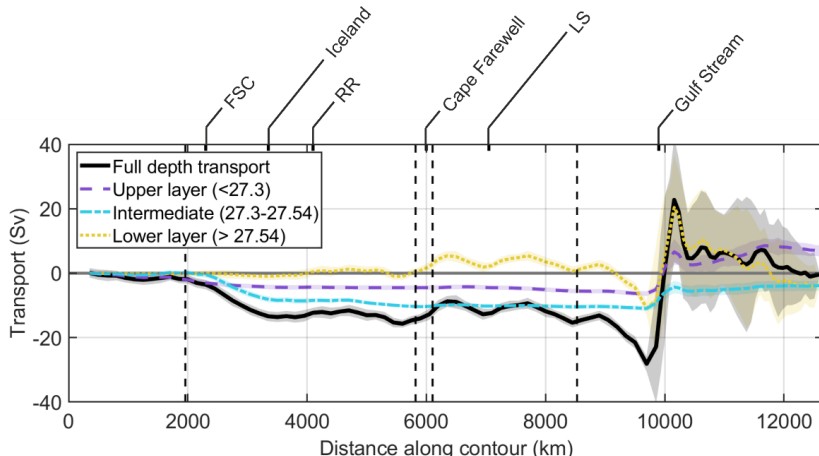

*Figure 8: Cumulative volume transport into the SPG interior (geostrophic + surface Ekman), between surface and seabed. Adjustment velocity applied below 1000 m to conserve volume. Transport in upper, lower and*
*intermediate layers also shown; these are defined later in Sect. 3.4. Key locations around boundary labelled as for Fig. 3, vertical dashed lines denote OSNAP crossings.*

The overturning stream function, ψ, is a measure of the amount of water transformed to higher densities in each density class. We compute the overturning in density space following Lozier et al., (2019):

$$\psi = \int_{\sigma_{min}}^{\sigma} \int_{x_{end}}^{x_{start}} v \frac{\partial z}{\partial \sigma} dx d\sigma \ [Sv] \tag{9}$$

This is shown for each season, and for the annual mean, in Fig. 9a. The main peak in the overturning stream function occurs at densities between 27.26 and 27.30 kgm⁻³ (Fig. 9, Table 2), with maximum overturning varying between 6.20 Sv in summer and 10.17 Sv in spring. A smaller secondary peak exists at higher density classes (27.54 to 27.58 kgm⁻³) in all seasons but winter, with maximum overturning values of 3.59 to 5.50 Sv.

We investigate this signal by deconstructing the mean (Figs. 9b and c). Figures 9b and c show the transports
accumulated over density space for the 1000 m isobath and 47° N transect components of our section





respectively. About 8 Sv is exported across the 1000 m contour in the density range 27.3 kgm$^{-3}$ to 27.42 kgm$^{-3}$ (Fig. 9b). The maximum density encountered on the boundary contour is 27.74 kgm$^{-3}$, with net transports of -17 to -25 Sv. Across 47° N, there is a steady accumulation of density over all density ranges between 26.9 kgm$^{-3}$ and 27.54 kgm$^{-3}$, with about 18 Sv accumulated.

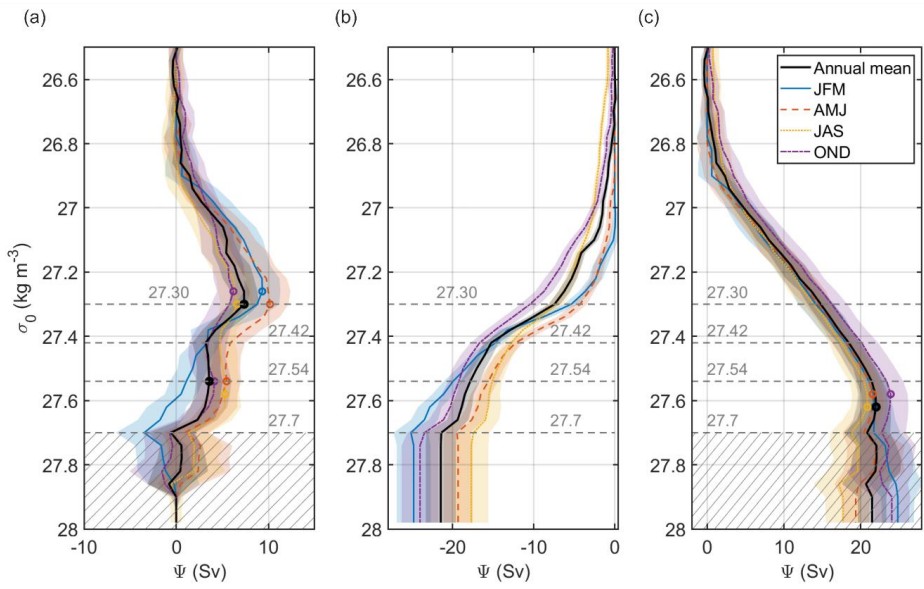


*Figure 9: (a) Overturning stream function ψ for full SPG boundary in density space between surface and seabed using corrected velocities for sub-1000 m currents. Density of maximum overturning, and that of secondary peak where applicable, highlighted by circles. Densities of mean inflection points marked by horizontal grey dashed lines; these are overlaid on Fig. 3d. The hatched area denotes the approximate density space impacted*

*by the sub-1000 m correction velocities (see Supplementary Materials S5 for details). (b) same, but for boundary contour (0-9500 km) only, (c) same, but for 47° N transect (9500 – 12700 km) only.*



*Table 2: Overturning strength and its location in density space, by season.*

|  | **Max overturning (Sv)** | **Isopycnal of maximum overturning (kgm$^{-3}$)** |
|---|---|---|
| **Annual mean** | 7.36 ± 1.48 | 27.30 |
| **JFM** | 9.33 ± 2.02 | 27.26 |
| **AMJ** | 10.17 ± 1.91 | 27.30 |
| **JAS** | 6.59 ± 1.35 | 27.30 |
| **OND** | 6.20 ± 1.40 | 27.26 |

**3.5. Heat and freshwater fluxes between the Subpolar Gyre and the boundary**

**3.5.1. Advective fluxes**

The SPG on average gains heat of 0.18 ± 0.05 PW via advection (Fig. 10, Table 3). 0.25 PW is exported in the region of the FSC and 0.4 PW gained across 47° N mainly in the Gulf Stream/NAC. Over much of the boundary little heat is exchanged with the exterior, because the temperatures are close to the reference temperature $\bar{\theta}$ (4.03 °C, Eq. (5)). There is some heat loss to the exterior between 0 and 1000 km due to outflow combined with above average temperatures. There is very little seasonality in heat flux across the 1000 m contour (0-9500 km).

Advection drives a net salinification of the SPG, with a net freshwater loss of -0.10 Sv. Freshwater flux is largely into the SPG up to 6500 km, driven by the net export of waters with salinity higher than the reference salinity. The NAC is responsible for the effective gain of 0.1 Sv of freshwater due to this effect. As for heat flux, there is little seasonality in freshwater flux around the boundary. An exception is off south-west Greenland, where fresher upper waters during the winter (Fig. S2), in conjunction with increased surface Ekman

transport (Fig. 6) do result in a localised seasonal gain of 0.02 Sv. Between the western Labrador Sea and the Flemish Cap about 0.08 Sv of freshwater is exported from the SPG before reaching 47° N. The effective negative freshwater flux of the Gulf Stream (-0.1 Sv) is the result of a positive volume flux associated with water of higher salinity than the basin-mean ($\bar{S}$, Eq. (6)).

The local heat and freshwater fluxes and their signs depend on the reference values $\bar{\theta}$ and $\bar{S}$ used (Eq. (5) and

(6)). For heat flux we use the mean temperature of the waters of the full-depth SPG interior enclosed by the boundary (4.03 °C). The heat fluxes thus have a physical meaning in that they show the level to which these waters warm or cool the SPG. Similarly, for freshwater flux we use the mean salinity of the full-depth SPG interior (35.14 g kg$^{-3}$), thus showing the level to which the boundary fluxes freshen or salinize the SPG. As the flux calculations use mass-balanced velocities, the net heat fluxes into the SPG (Table 3) are insensitive to the

choice of reference temperature (Eq. (5)), but the net freshwater fluxes retain some sensitivity to the reference salinity due to the denominator of Eq. (6).



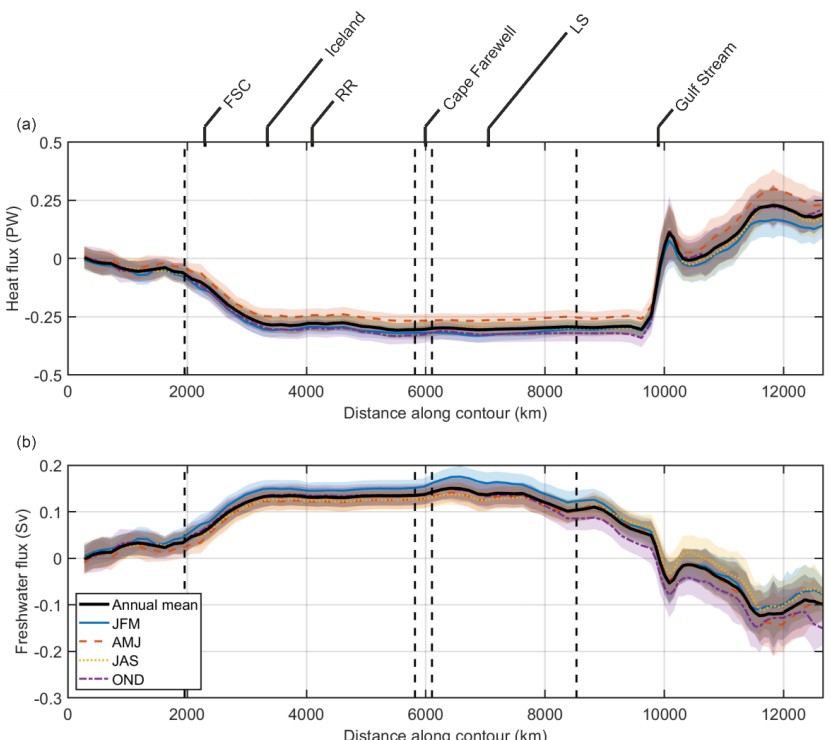

*Figure 10: Cumulative heat [PW] and fresh water [Sv] fluxes into the SPG between surface and seabed for (a) heat and (b) freshwater, using corrected velocities for sub-1000 m currents. Key locations around boundary labelled as for Fig. 3, vertical dashed lines denote OSNAP crossings.*

Spatially integrated (net) advective heat and freshwater fluxes into the SPG are shown in Table 3. Heat fluxes into the SPG range from 0.14 PW in winter to 0.23 PW in spring. Freshwater fluxes are negative in all seasons and are between –0.07 Sv (summer) and –0.15 Sv (autumn).

### 3.5.2. Surface heat and freshwater fluxes

In Table 3 we also show the seasonal and annual mean surface heat and freshwater fluxes derived from ERA5. The seasonal range of surface heat fluxes is much larger than that of the advective fluxes; between -0.80 PW (lost to atmosphere) in the winter and 0.33 PW (gained from atmosphere) in the spring. The annual mean surface heat loss (-0.24 PW) is of a similar magnitude to the advective heat flux into the SPG. Seasonality in freshwater surface fluxes are weak, ranging from 0.05 Sv in winter and spring to 0.08 Sv in the summer with an annual mean of 0.06 Sv into the SPG.



*Table 3: Net fluxes into SPG [between surface and seabed]*

|  | Heat flux (PW) | | Freshwater flux (Sv) | |
|---|---|---|---|---|
|  | Advective flux | Downward surface Flux (ERA5) | Advective flux | Downward surface flux (ERA5) |
| **Annual mean** | 0.18 ± 0.05 | -0.24 ± 0.02 | -0.10 ± 0.03 | 0.06 ± 0.01 |
| **JFM** | 0.14 ± 0.05 | -0.80 ± 0.04 | -0.08 ± 0.02 | 0.05 ± 0.02 |
| **AMJ** | 0.23 ± 0.05 | 0.33 ± 0.02 | -0.10 ± 0.02 | 0.05 ± 0.01 |
| **JAS** | 0.17 ± 0.05 | 0.27 ± 0.02 | -0.07 ± 0.02 | 0.08 ± 0.01 |
| **OND** | 0.21 ± 0.05 | -0.77 ± 0.03 | -0.15 ± 0.02 | 0.06 ± 0.01 |

### 3.5.3. Boundary topography and its relationship with turbulent eddy fluxes

Steeply sloping margins are known to be rich in eddy activity (Spall and Pickart 2000, Brüggemann and Katsman, 2019). It is conceivable therefore that eddy exchange of heat and freshwater may be significant contributors to SPG-boundary exchange. The slope angle of the SPG boundary and its relationship to the EKE along the 1000 m contour is shown in Fig. 11.

The spatial distribution of EKE along the 1000m contour appears relatively consistent between seasons but during the autumn and winter EKE is about double that of spring and summer. EKE is greatest around Greenland and in the western Labrador Sea during all seasons, with the WGC values exceptionally high. A similar spatial structure emerges when examining the slope angle around the SPG boundary, with Fig. 11 showing the excellent agreement between the two parameters. Note that the extreme (>=20 °) slope west of Greenland corresponds to the EKE maximum in the WGC.





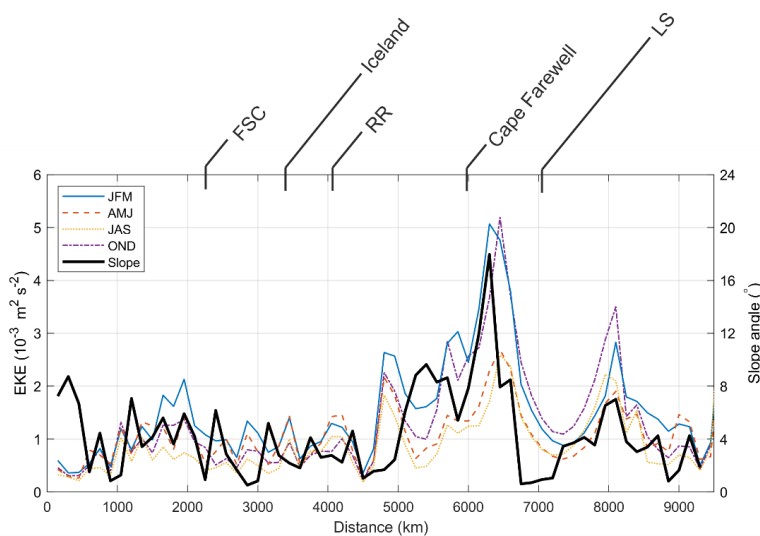

*Figure 11: Angle of continental slope (black) compared to EKE by region. Key locations around boundary labelled as for Fig. 3, note x-axis excludes 47° N transect.*

An estimate of the diffusive heat flux associated with eddy activity was made using satellite derived SST and surface geostrophic velocities and is detailed in Supplementary Materials S4. Heat is diffused out of the SPG along the 1000 m contour, and into the SPG along 47° N (Fig. S3). A total of 0.0062 PW of heat energy enters the SPG via turbulent diffusion, roughly two orders of magnitude less than the contribution from advection so this process is not included in our heat budget. It was not possible to estimate diffusive freshwater flux due to the lack of reliable satellite SSS observations.

### 4. Discussion

In this article we present the first comprehensive observational assessment of properties, transports and fluxes between the interior and exterior of the whole North Atlantic SPG. In conjunction with model data, we used this to identify the relative importance of processes driving fluxes across the boundary. Our observation-based approach uses data from 2000 to 2019, so can be considered the present mean state of circulation on decadal timescales. By considering fluxes into and out of the SPG as a whole, this work provides a measure of which processes in the SPG interior contribute to the AMOC.

#### 4.1. Overturning in the Subpolar Gyre

Here we discuss the overturning stream function for the boundary of the SPG, and what it implies for water transformation within the SPG.

We found the maximum of the annual mean overturning streamfunction in density space to be $7.36 \pm 1.48$ Sv across the 27.30 kgm$^{-3}$ isopycnal (Table 2, Fig. 9a). To contextualise this value, the mean overturning measured across the 26° N zonal section by the RAPID-MOCHA array is 16.8 Sv (Smeed et al., 2018), while mean





overturning across the ~60° N OSNAP section, which bisects the SPG, is 16.6 Sv (Li et al., 2021a). Other estimates in the North Atlantic place the total overturning between 11.9-18.4 Sv (Caínzos et al., 2022; Fraser and Cunningham, 2021; Rossby et al., 2017; Sarafanov et al., 2012). However, estimates from transatlantic zonal sections only speak to water mass transformation north of the line in question. Our overturning stream function (Fig. 9a) represents the water mass transformation in the interior of the SPG, an enclosed volume, and is therefore a measure of overturning within the SPG volume.

Comparisons with other regionally bounded estimates of SPNA overturning are useful for interpreting our results dynamically. For example, our estimate for overturning is very similar to the overturning between OSNAP-East and the GSR estimated by Petit et al., (2020) (7.0 Sv). From this, one might infer that virtually all the overturning in the SPG happens in the Irminger and Iceland Basins. However, these estimates are not directly comparable for two reasons. Firstly, the domain covered by Petit et al., (2020) includes shallow and coastal regions comprising turbulent boundary currents and air-sea incteractions over the East Greenland shelf, Reykjanes Ridge and GSR, all of which which are outside our domain. Secondly, Petit et al., (2020) estimate overturning in the interior by subtracting AMOC strength at the GSR from the AMOC strength at OSNAP east, although these two overturning maxima do not necessarily coincide in density space, which may result in an underestimate of the actual overturning taking place in the region. On the other hand, our study finds a maximum overturning strength of 7.36 Sv across a common isopycnal, so the class of water mass transformation being quantified is consistent around the boundary. This is analogous to subtracting the overturning streamfunction at the GSR from the overturning streamfunction at OSNAP east and taking the maximum value of the residual. While either approach might be considered a measure of overturning, the resulting values have different dynamical implications and are not directly comparable.

Our results reveal that the peak of the water mass transformation processes within the SPG occurs across the 27.30 kgm$^{-3}$ isopycnal. The net inflow at 47° N (Fig. 9c) is evenly distributed across a wide density range (26.9 < $\sigma_0$ < 27.54 kgm$^{-3}$), while the net outflow across the 1000 m isobath is concentrated around 27.35 kgm$^{-3}$. The overturning maximum at $\sigma_0$ = 27.30 kgm$^{-3}$ therefore corresponds to the transformation of 7.36 Sv of upper water ($\sigma_0$ < 27.3 kgm$^{-3}$) which enters the SPG from the south before being cooled by the atmosphere to form SPMW (27.3-27.54 kgm$^{-3}$). Around half (3.77 Sv) of this SPMW is then exported from the SPG (Fig. 9a). The transports by density layer (Fig. 8) reveal that the outflow in this (intermediate) density class is located between Iceland and Scotland, as it is carried northwards in the NAC.

The remaining SPMW is further transformed within the SPG, resulting in the broad plateau in the mean overturning between 27.4 to 27.6 kgm$^{-3}$ with a secondary overturning maximum at $\sigma_0$ = 27.54 kgm$^{-3}$. This density range corresponds to isopycnals outcropping in the Irminger and Labrador Seas (e.g. Lozier et al., 2019), indicating that this secondary transformation occurs as the remaining SPMW circulates into the western SPG. The resulting dense water ($\sigma_0$ > 27.54 kgm$^{-3}$) is then exported both via the Labrador Current and across 47° N (Fig. 8). Dense water also enters the SPG both at Cape Farewell, having presumably travelled south through the Denmark Strait, and in the Gulf Stream, which partially cancels the outflow at 47° N. However, the net export in this layer indicates dense water is formed, at a rate of 3.59 Sv, in the SPG interior.



The overturning maxima at $\sigma_0 = 27.30$ kgm$^{-3}$ and $\sigma_0 = 27.54$ kgm$^{-3}$ are both much lighter than the isopycnal of maximum overturning reported for OSNAP (27.66 kgm$^{-3}$, Lozier et al., 2019) and are instead comparable with the outcropping isopycnals implicated in SPMW formation (27.3-27.5 kgm$^{-3}$, Petit et al., 2021). This is because the GSR overflows which dominate the lower limb transport at OSNAP east are formed outside of the SPG, and therefore contribute minimally to the overturning structure computed around our closed-loop boundary. The negative overturning values during autumn and winter (Fig. 9a) indicate that these overflow waters can become

lighter inside the SPG. This modification may result from deep winter mixing making the dense overflow water part of the surface layer, which is lighter than the overflow water.

The deepest overflows ($\sigma_0 > 27.7$ kgm$^{-3}$) are not resolved by the boundary climatology (Fig. 9b). However, and in reality, these waters will flow southward through the SPG at depth with little exposure to the atmosphere. They therefore undergo minimal transformation within the SPG so their inclusion would not significantly alter

the structure of overturning we observe (Fig. 9a). We address the issue of deep overflows in Sect. 4.5.

The maximum overturning at $\sigma_0 \approx 27.30$ kgm$^{-3}$ has significant seasonal variability (Fig. 9a), with substantially larger values in winter and spring (9.33 Sv, 10.17 Sv) than in summer and autumn (6.59 Sv, 6.20 Sv). This is in accord with the seasonal overturning cycle now apparent north of OSNAP, such that overturning lags the winter surface cooling maximum by one season (Li et al., 2021a; Petit et al., 2020; Petit et al., 2021). This suggests that

newly-formed SPMW is exported via the NAC with a lag time of a few months. In our SPG volume budget, the pronounced seasonality of surface Ekman transport out of the SPG is the main driver of seasonal differences (Fig. 6). However, the one-season lag between peak surface Ekman transport and peak overturning shows that pressure/density differences must also play a role.

The secondary maximum in the overturning at $\sigma_0 \approx 27.54$ kgm$^{-3}$ displays a different class of seasonal variability

(Fig. 9a). The transformation is strongest in spring and summer (5.46 Sv, 5.31 Sv), while the autumn value (4.13 Sv) is close to the mean (3.59 Sv). In winter this secondary peak is absent, as virtually all the SPMW formed in the eastern SPG is exported before undergoing further transformation to dense water ($\sigma_0 > 27.54$ kgm$^{-3}$). This may be due to the strong surface Ekman component in winter (Fig. 6) driving export of SPMW onto the shelf around the western Irminger and Labrador basins.

In summary, we see 7.36 Sv net import of upper waters ($\sigma_0 < 27.30$ kgm$^{-3}$) which are transformed in the interior then exported, in approximately equal measure, as either intermediate water (27.30-27.54 kgm$^{-3}$) in the NAC or as dense water ($\sigma_0 > 27.54$ kgm$^{-3}$) exiting to the south. These results support the findings of Petit et al., (2021); that the pre-conditioning of buoyant NAC waters into SPMW is a key stage in the transformation of water to successively higher densities and that it is therefore an important driver and modulator of AMOC strength.

**4.2. Heat and freshwater divergence in the Subpolar Gyre**

We find a net advective convergence of heat into the SPG of $0.18 \pm 0.05$ and a net divergence of freshwater of -$0.10 \pm 0.03$ (Table 3). Are these compatible with atmospheric fluxes?

The annual mean net downward heat flux over the SPG is $-0.24 \pm 0.02$ PW (Table 3). Thus, our estimate for the mean heat imported into the SPG through advection is approximately balanced by the mean loss to the

atmosphere. The seasonal range of surface heat fluxes (-0.80 PW in the winter to 0.33 PW in the spring) is





much greater than that for advective heat fluxes (0.14 PW in the winter to 0.23 PW in the spring). The annual mean net downward freshwater flux is $0.06 \pm 0.01$ Sv with only minor seasonality.

A discrepancy of -0.06 PW remains between the rate of heat entering the SPG through advection and that of heat leaving the SPG through surface cooling averaged over 20 years. This value is compatible with the
observed magnitude of cooling in the North Atlantic, for example Bryden et al., (2020) find cooling at rate of 0.04 PW for the region 26-70° N between 2008-2016. For freshwater however, the discrepancy between the rates of advective freshwater export and surface freshwater import (-0.04 Sv) implies a net salinification during the period 2000-2019 which is contrary to the findings of Bryden et al., (2020), who reported freshwater loss at a rate of 0.062 Sv for the region 26-70° N between 2008-2016. We note that for both heat and freshwater
fluxes, the discrepancy is within our error bounds so cannot be significantly distinguished from zero.

We find that a mean of 0.48 PW crosses 47° N into the SPG (Fig. 10a). This is not directly comparable to other zonal transects (Fraser and Cunningham 2021; Li et al., 2021b; Lozier et al., 2019) as our domain does not extend to the coast. However, we can estimate that if 0.48 PW is transported into the interior in the south, and 0.18 PW is lost in the SPG then 0.30 PW exits the SPG across the 1000 m contour. Similarly, a freshwater flux
of -0.15 Sv across 47° N (Fig. 10b) and a divergence of -0.10 Sv across the SPG implies a freshwater transport of 0.05 Sv across the 1000 m contour.

The heat entering via the Gulf Stream (0.3 PW) reduces to 0.25 PW exiting via the FSC, suggesting that the NAC loses 0.05 PW of heat in the SPG. Similarly, the NAC gains freshwater at a rate of 0.01 Sv in the SPG. It is interesting to note that substantial heat and freshwater is exchanged with the boundary south of the FSC (in
the Rockall Trough), driving warming / salinification of the slope current, the NW European shelf and the North Sea.

The contribution of the energetic EGC and WGC systems to the overall SPG heat and freshwater budgets is relatively small. While the region around Greenland contributes up to 0.02 Sv of freshwater, the melting from the Greenland ice sheet appears to play a minor role in the freshwater budget of the SPG. This may be because
much of the freshwater remains on the shelf rather than joining the EGC (De Steur et al., 2009). Near Cape Farewell, the ingress of 8 Sv of relatively dense water signals the import of various modified water masses across the Denmark Strait, entering the SPG chiefly through the EGC and WGC. The precise location of import is dependent on how our boundary intersects with the EGC and WGC cores, but the accumulated fluxes are robust to this effect. Net heat flux resulting from this interface is minimal because local temperatures are near
the reference temperature $\bar{\theta}$ (Fig. 10a, Eq. (5)).

While turbulent diffusion does not play a significant role in the SPG heat budget, the highly energetic Gulf Stream eddy field does import 0.025 PW, or about 8 % of the total Gulf Stream heat input, though this is largely compensated by heat leaving the SPG on either side (Fig. S3). This value is still an order of magnitude smaller than the eddy heat flux estimated at 36° N (immediately after its separation at Cape Hatteras) by Tréguier et al.,
(2017) (0.3 PW), though this is perhaps unsurprising given the profound change in the character of the Gulf Stream between 36° N and 47° N.



### 4.3. Buoyancy exchanges in the western Subpolar Gyre

It is clear from Fig. 3 that that density generally increases with progress around the SPG boundary, and that this is primarily caused by gradual cooling. This reflects the buoyancy loss in the interior and is also seen in the

boundary current as it flows around the basin (Straneo, 2006). A notable exception to the increasing density trend is south-western Greenland where an injection of freshwater (and warming below 250 m) leads to a marked reduction in density, along with a reversal of volume transports between the interior of the SPG and the boundary in this region (Fig. 5). While Liu et al., (2022) diagnose upwelling in this region, in our climatology this positive geostrophic transport appears to be due to the movement of the boundary current relative to the

1000 m contour. We note that the reversal of the prevailing horizontal density gradient is a subtlety that is lacking from more idealised studies of boundary current dynamics which assume a continual decrease of density with progress counter-clockwise around the SPG (e.g. Brüggemann and Katsman, 2019).

The role of the EGC and WGC in water mass modification is undoubtedly enhanced by eddy activity, which is in part driven by the boundary current interacting with local topography. In Fig. 11 we show that the

exceptionally high EKE values in the WGC region are associated with a slope of 20° west of Greenland. This coherence suggests that EKE, and hence the diffusive flux of buoyancy between the boundary and SPG, is controlled by the steepness of the sloping margins. As we have already stated, diffusive fluxes have minimal significance for the overall boundary heat and freshwater budget, but in the WGC region we find that diffusive and advective fluxes are comparable. Studies citing eddy diffusion for communication between interior and

boundary tend to be located around southern Greenland (Brüggemann and Katsman, 2019; Le Bras et al., 2020; Liu et al., 2022) and our results highlight that this region is an exception to the rule around the gyre.

We do not see clear signs of true winter deep convection at the boundary of the Labrador Sea (Fig. S2, mixed layer depths of > 800 m are indicated by Lavender et al., 2000). Deep convection may be largely confined to the basin interior in the winter, communicated to the boundary and appearing as anomalies at the boundary in

spring (Yashayaev and Loder, 2017, Fig. S2).

### 4.4. Subpolar Gyre volume budget estimation

Given the transports estimated in this study, we can make a first order estimate of the SPG volume budget given the continuity constraint of zero net transport.

The SPG interior can be divided into an upper and lower volume partitioned at 1000 m depth. The upper

volume is enclosed by the 1000 m boundary curtain and the 47° N section, the lower volume is completely enclosed except across 47° N (Fig. 12). A net inflow above 1000 m must be balanced by downwelling across the 1000 m 'surface'. In addition, model-based estimates of North Atlantic AMOC in depth space find maximum overturning located near 1000 m (Biastoch et al., 2021; Hirschi et al., 2020) so the vertical transport across 1000 m is approximately equivalent to the strength of the AMOC in the SPG. This balance is depicted in

Eq. (10):

$$Q_{inflow < 1000\,m} = Q_{outflow > 1000\,m\ on\ 47°\,N} = W_{downwelling\ through\ 1000\,m} \approx AMOC_z \qquad (10)$$



The $Q_{inflow<1000\,m}$ term can be expressed as the sum of the geostrophic, surface Ekman and bottom Ekman transports, with a remainder term necessary to capture flows not resolved by the observational budget:

$$Q_{inflow<1000\,m} = Q_{geo} + Q_{Ek\,surf} + Q_{Ek\,bed} + Q_{remainder} \tag{11}$$

Net geostrophic flow above 1000 m is only permitted due to the beta effect and is therefore constrained (mean +2.3 Sv). Note that this would be the case even if the hydrography were perfectly known. For surface Ekman we take the annual mean calculated from observations (-4.9 Sv, Sect. 3.2.2). For bottom Ekman we use the estimate from Argo trajectories, using a mean velocity of 11 cm s$^{-1}$ (+2.4 Sv, Sect. 3.2.3). Given the approximate cancellation of the mean geostrophic, surface Ekman and bottom Ekman terms (totalling -0.2 Sv),

$Q_{remainder}$ is left as the dominant term on the RHS. Thus on average, almost all the southward flow below 1000m at 47° N is driven by the $Q_{remainder}$ term (note that the sum of geostrophic, surface Ekman and bottom Ekman terms is seasonal (+2.25 Sv in summer, -3.0 Sv in winter) with the seasonality driven by the surface Ekman term (Table 1)). We might therefore state that on average,

$$Q_{inflow<1000\,m} = Q_{outflow>1000\,m\,on\,47°\,N} = W_{downwelling\,through\,1000\,m} \approx Q_{remainder} \tag{12}$$

Hence from Eq. (10) and (12),

$$Q_{remainder} \approx AMOC_z \tag{13}$$

We estimate the mean $Q_{remainder}$ to be 12.0 Sv using VIKING20X (dashed grey line, Fig. 7b). From Eq. (11), this results in a net gain of +11.8 Sv above 1000 m, necessitating a downwelling flow of 11.8 Sv through the 1000 m surface, and an equivalent southward net flow across 47° N below 1000 m (Fig. 12). The depth space

AMOC estimated by Hirschi et al., (2020) and Biastoch et al., (2021) is 10-15 Sv and therefore is of the same order as that inferred from the VIKING20X remainder term.



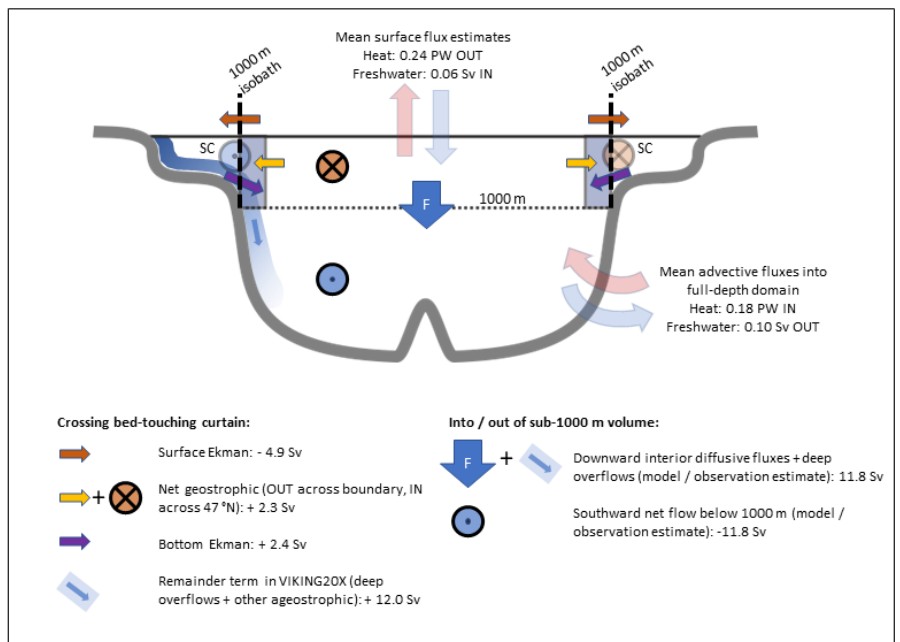

*Figure 12: Schematic of SPG boundary and interior processes contributing to transport through the SPG,*
*viewed from 47° N section. The shaded rectangles on either side of the basin represent the regions in which*
*CTD data were gathered. SC: Slope current, Net flow across 47 N above 1000 m is northward (into the SPG)*
*and below 1000 m it is southward (out of the SPG). A net downwelling (F) is required to balance the transports*
*in and out of the SPG.*

**4.5. The role of unresolved flows**

Between the UK and Greenland (Fig. 5) 12 Sv of geostrophic transport leaves the SPG.   This is the same
transport as was reported for the upper limb across OSNAP-East (Lozier et al., 2019) and implies that the return
current in the lower limb is not captured in the geostrophic transports from the observational analysis.  Another
indication that the lower limb of the AMOC is not fully resolved in the observations is the lack of very cold ($<$ 3
°C) and dense ($>$ 27.8 kgm$^{-3}$) waters where we would expect the Faroe Bank Channel overflow and Denmark
Strait Overflow (DSO) to bisect the boundary (Johnson et al., 2017; Mastropole et al., 2017).  The dominant role
of $Q_{remainder}$  in Sect. 4.4 further highlights that some processes are not fully captured by the observational
analysis.  In this section we consider which regions and dynamical processes contribute to the $Q_{remainder}$ term.

The region off south-east Greenland is responsible for over half the $Q_{remainder}$ signal in VIKING20X (Fig. 7).
We surmise that the modelled DSO is primarily responsible for this transport.  The model fields suggest a
contribution by $Q_{remainder}$ of 6.0-6.8 Sv entering the SPG in the Denmark Strait region, with the majority of the
flow close to the seabed (dashed grey line, Fig. 7).  For comparison, observational estimates of the volume
transport in the overflow suggest that the DSO is responsible for 2.5-3.8 Sv (Girton et al., 2001; Jochumsen et



al., 2012; Käse et al., 2003). A possible explanation for the 'remainder' term in VIKING20X overestimating the DSO is an upslope geostrophic component which acts to reduce the net transport in this region (Fig. 7b).

We have encountered modelling results suggesting that the DSO may have a significant ageostrophic and non-Ekman component and must therefore receive significant contributions from non-linear and viscous processes. The DSO is manifest as a turbulent cascade released over the sill in pulses with a timescale of around 3-5 days (e.g. Käse et al., 2003). One would anticipate that small-scale, non-linear and ageostrophic processes would dominate in such an environment. It is beyond the scope of this paper to quantitively assess these processes.

However, our analysis demonstrates the importance of overflow dynamics in closing the overturning streamlines in the SPNA.

There are several reasons why our sampling strategy and analysis may result in a poorly resolved DSO. Firstly, the profiles contributing to our dataset may on average be too far from the continental slope to regularly capture the overflow (see schematic in Fig. 12). Secondly, due to the transitory nature of the DSO temporally-scattered

CTD sampling may fail to sample it. Finally, Argo floats may be actively deflected around the downslope-flowing boluses of dense water, thus not sampling the core properties.

**5. Conclusions**

A novel observational climatology of the entire SPG boundary has yielded new perspectives on overturning in the interior of the SPG. We find an average transformation of $7.36 \pm 1.48$ Sv of upper waters ($\sigma_0 < 27.30$ kgm$^{-3}$)

occurs within the SPG with a seasonal maximum in spring and minimum in the autumn, lagging surface buoyancy forcing by one season. The products of upper water transformation are intermediate water (27.30-27.54 kgm$^{-3}$) exiting in the NAC or dense water ($\sigma_0 > 27.54$ kgm$^{-3}$) exiting to the south. These findings underline the findings of Petit et al., (2021): that the overturning of dense waters is reliant on the prior 'pre-conditioning' of lighter waters.

We find a mean advective convergence of heat into the SPG of $0.18 \pm 0.05$ PW, and a net divergence of freshwater of $-0.10 \pm 0.02$ Sv, which are approximately balanced by surface fluxes. Net diffusive heat and freshwater fluxes into the SPG are negligible, but hotspots of eddy activity such as the Gulf Stream and western Greenland result in localised diffusive heat fluxes approaching those of the advective contributions.

When considering the total transports into and out of the SPG volume, we find that the mean geostrophic (2.3

Sv), surface Ekman (-4.9 Sv) and bottom Ekman (2.4 Sv) terms approximately cancel, meaning that flow downwards across the 1000 m surface is dominated by ageostrophic (and non-Ekman) processes. This result highlights the requirement to better understand the overflows into the SPG and demonstrates that a geostrophic approach alone may not be sufficient for this.

Our investigation focused on the recent (20-year) climatic mean state, as was necessitated by observational data

availability. However, given recent evidence of changes in large-scale circulation patterns (Biastoch et al., 2021; Fox et al., 2022; Zhang and Thomas, 2021) it is crucial to assess the decadal shifts in the basin-scale processes outlined here, and establish to what extent this can alter the behaviour of the AMOC.



**6. Data and code availability**

Aggregated Argo and CTD profile data are available from the WOD at https://www.ncei.noaa.gov/access/world-
ocean-database-select/dbsearch.html. Gridded EN4 observations data can be obtained from
https://www.metoffice.gov.uk/hadobs/en4/download-en4-2-2.html. ECMWF ERA5 reanalysis data is available
at https://cds.climate.copernicus.eu/cdsapp#!/dataset/reanalysis-era5-single-levels. AVISO sea surface height
data can be obtained from the CMEMS portal: https://resources.marine.copernicus.eu/product-
detail/SEALEVEL_GLO_PHY_L4_MY_008_047/INFORMATION. GEBCO bathymetry data can be
downloaded from https://www.gebco.net/. The code for computing fluxes and overturning from the boundary
climatology is available at https://github.com/sjones1000/SPG_boundary. The NEMO code used in
VIKING20X is available at https://forge.ipsl.jussieu.fr/nemo/svn/NEMO/releases/release-3.6 (NEMO System
Team, 2021). Our experiments are based on revision 6721. The original underlying VIKING20X model output
is available on request from GEOMAR research data management (datamanagement@geomar.de).

**7. Author contribution**

SJ conducted the core analyses and prepared the manuscript with contributions from all co-authors. NF
computed the surface Ekman transports and EKE values around the boundary and contributed to manuscript
preparation and interpretation of results. SC secured the funding for the work, conceptualised the gyre boundary
investigation, and contributed to the text and experiment design throughout. AF supplied the VIKING20X
investigation and contributed to the text and interpretation of results. MI supplied the slope angle investigation
and contributed to the text and interpretation of results.

**8. Competing interests**

The authors declare that they have no conflict of interest.

**9. Acknowledgements**

This project was supported by the NERC research programme funding CLASS NE/R015953/1, OSNAP
NE/K010700/1 and SNAP-DRAGON NE/T013494/1 (N. Fraser and S. Cunningham). Additional support (AF)
was received from the European Union Horizon 2020 research and innovation program under grant 727852
(Blue-Action). This output reflects only the author's view and the European Union cannot be held responsible
for any use that may be made of the information contained therein.

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
