# Peer review of "Observation-based estimates of volume, heat and freshwater exchanges between the subpolar North Atlantic interior, its boundary currents and the atmosphere"

_EGUsphere, 2022_

## Referee Comment (RC1)

**Observation-based estimates of volume, heat and freshwater exchanges between the subpolar North Atlantic interior, its boundary currents and the atmosphere**
by Sam C. Jones, Neil J. Fraser, Stuart A. Cunningham, Alan D. Fox, and Mark E. Inall

This manuscript presents an innovative analysis of recent North Atlantic hydrographic measurements along with ancillary data sets to quantify the overturning in the subpolar gyre.

I think this is a well-written and very interesting paper that significantly contributes to the understanding of water mass transformation in the subpolar North Atlantic. In my opinion, the analysis has two major weaknesses, namely (i) that the perimeter contour for large parts of the domain is oriented along the major boundary current system such that the cross-contour component of the flow is a small residual relative to the along-contour flow and (ii) that the perimeter contour has too coarse resolution to capture important components of the circulation, in particular the overflows through Denmark Strait and Faroe Bank Channel. I am not sure that these two weaknesses can be robustly addressed, but I think that at least a more extensive discussion of these two concerns is necessary. I also have a few other comments that I hope the authors will consider. Finally, I would like to emphasize that I think this manuscript ingeniously utilizes existing observations to address an important and challenging scientific question; I support the effort and encourage the authors to submit a revised version of the manuscript.

**General comments:**

Most of the major currents of the subpolar gyre boundary current system have substantial flow along the 1000 m isobath, such as the East Greenland (Le Bras *et al.*, 2020) and West Greenland Currents (Pacini *et al.*, 2020), and the dominant isopycnal slope along this depth contour is across the boundary current system. These currents are to varying extent subject to meanders and instabilities (e.g., Prater, 2002; Pacini and Pickart, 2022), which introduce substantial variability in the measurements. Quantifying the cross-slope geostrophic flow from the isopycnal slope along a depth contour that is characterized by substantial and vigorous dynamics is not optimal. Choosing a deeper isobath for the perimeter contour would likely alleviate the problem. While there are other good reasons for choosing the 1000 m depth contour, I think more robust estimates of cross-contour flow would be obtained along a deeper isobath. I will not advocate that the analysis is redone, but would like to see this issue more extensively discussed in the paper.

For the horizontal gridding along the perimeter contour, a resolution of 150 km was used. The coarse resolution may suffice for the large-scale, geostrophic interior-boundary exchange. However, very important contributions to this exchange occurs on much smaller spatial scales, in particular eddies and deep overflows from the Nordic Seas, but also currents such as the Deep Western Boundary Current are not resolved at this scale. While eddies may roughly balance in- and outward fluxes across the perimeter contour, the overflows and the water masses they entrain are crucial inflows into the subpolar gyre that will not have been properly accounted for. Downstream of Denmark Strait, the overflow plume has a spatial scale of much less than 100 km and rapidly descends beneath the 1000 m depth contour (Dickson and Brown, 1994; Girton and Sanford, 2003) – hence the Denmark Strait overflow cannot be the main source of inflow near Cape Farewell.

**Specific comments:**

Line 25:
Is the minimum overturning in fall mainly comprised of the overflows from the Nordic Seas, which would form a steady baseline with minimal seasonal variability, while the maximum in spring also includes dense water formed within the subpolar gyre? Or is the surface Ekman component also an important source of variability? I think it would be good to specify the cause of this seasonal variability already in the abstract.

Line 36:
Including the estimate of 305±26 TW across the Greenland-Scotland Ridge by Tsubouchi *et al.* (2021) would be another very relevant point of comparison here.

Line 48:
Another important component of the return flow from the Arctic Ocean and Nordic Seas is the surface outflow of Polar Water in the East Greenland Current (de Steur *et al.*, 2017).

Line 60:
The relatively low impact of water mass transformation in the Labrador Sea on the AMOC was known also prior to OSNAP (e.g., Pickart and Spall, 2007).

Line 64:
The importance of water mass transformation north of the Greenland-Scotland Ridge for supply of dense water to the lower limb of the AMOC should not be underestimated (e.g., Chafik and Rossby, 2019).

Line 99:
Another important dynamical consideration is that, apart from the overflows, most of the sinking occurs along the boundary (Spall and Pickart, 2000; Johnson *et al.*, 2019)

Line 148:
If the minimal search radius is 150 km, equal to the distance between grid points, most of the profiles are probably used in more than one grid cell. Did you apply any weighting to emphasize the contributions of profiles closer to the grid point or the perimeter contour?

Line 199:
This is a great approach to estimate the statistical uncertainty inherent in the data set. Providing the measurement errors that are also inherent in the data set, such that the magnitudes of the statistical and measurement uncertainties can directly be compared, would also be good.

Line 210:
Please provide some more details regarding the calculation of the surface Ekman transport. For example, what have you taken to be the depth of the Ekman layer?

Line 258:
Important flows on relatively small scale such as the overflows from the Nordic Seas and the East Greenland Spill Jet (Pickart *et al.*, 2005) are not properly resolved. Perhaps EN4 at 47°N is not the sole cause of the imbalance, if these features, along with the Deep Western Boundary Current, substantially contribute to the imbalance?

Line 283:
What are the length scales over which the satellite absolute dynamic topography product was smoothed? On line 183 it is stated that smoothing was applied to mimic the smoothing inherent in the hydrographic gridding

process. Is the resulting length scale of the eddy kinetic energy consistent with eddies scaled by the Rossby radius, or were all eddies, to the extent that they were represented in the raw satellite record, removed by the smoothing procedure?

Line 308:
Offshore fluxes of freshwater from the Greenland shelf into the interior Labrador Sea near Cape Farewell (Lin *et al.*, 2018) and farther north where the West Greenland Current encounters steep topography and becomes unstable (e.g., Fratantoni, 2001; Prater, 2002) are likely major contributors to the cold, fresh low-density layer. Both of these processes are primarily eddy-driven, hence postulating that a portion of the West Greenland Current crosses into the interior supbolar gyre may not be necessary. These processes will not be resolved at 150 km horizontal resolution. Given the turbulent nature of these fluxes, it is also not obvious that there will be a consistent geostrophic flux across the perimeter contour.

Lines 349 and 491:
Most of the Atlantic Water inflow from the subpolar gyre to the Nordic Seas takes place east of Iceland, roughly evenly split on either side of the Faroe Islands (Østerhus *et al.*, 2019). Given the course resolution of the perimeter contour, I think it is more appropriate to ascribe this flow to the Iceland-Scotland Ridge rather than the Wyville Thomson Ridge (note that Thomson is spelled without a p).

Line 351:
Note that there is also some flow of Atlantic Water northward through Denmark Strait (Jónsson and Valdimarsson, 2012; Semper *et al.*, 2022).

Line 352:
What is the magnitude of the retroflection of the East Greenland Current near Cape Farewell?

Line 357:
An export of 12 Sv from the subpolar gyre to the Labrador shelf is immense. Is this a realistic number? Could this be related to water sinking along the boundary (e.g., Johnson *et al.*, 2019) or merge with the boundary current system (a substantial portion of which appears to be inshore of the 1000 m isobath, Zantopp *et al.*, 2017)?

Line 450:
Good discussion of missing contributions. While the model has a much higher resolution, how confident can you be that it is able to realistically capture these features?

Lines 493, 631, and 696:
It is not obvious why there should be a lower layer inflow in the vicinity of Cape Farewell. This is too far south of Denmark Strait to be ascribed to the overflow (Dickson and Brown, 1994; Girton and Sanford, 2003), but perhaps the East Greenland Spill Jet (Pickart *et al.*, 2005) contributes? Please elaborate.

Line 525:
This estimate of advective flux across the Iceland-Scotland Ridge appears to be in reasonably good agreement with the estimate of Tsubouchi *et al.* (2021).

Lines 572 and 726:
There is substantial discussion of the high EKE west of Greenland in the literature (e.g., Fratantoni, 2001; Prater, 2002).

Line 599:

The overturning across the Greenland-Scotland Ridge would be another vital point of comparison (e.g., Østerhus *et al.*, 2019; Tsubouchi *et al.*, 2021).

Line 618:

How does the density surface of 27.30 kg/m$^3$ for maximum overturning compare to similar results from Lozier *et al.* (2019) and Petit *et al.* (2020)?

Line 623:

This is an important result, which substantially modifies the conclusions of Petit *et al.* (2020). Without velocity measurements, they considered this water mass transformation part of the overturning in the subpolar gyre, and concluded that more deep-water formation occurs in the subpolar gyre than in the Nordic Seas. You have demonstrated that a substantial portion of this intermediate-density water continues to the north, into the Nordic Seas, where it is further transformed. As such, densification in the subpolar gyre preconditions further water mass transformation in the Nordic Seas and is thereby important for the North Atlantic overturning, but it is not appropriate to ascribe that part of the water mass transformation to overturning in the subpolar gyre, since the water proceeds into the Nordic Seas in the upper layer rather than returning to the south at depth.

Line 639:

Overflow waters from the Nordic Seas become lighter as they mix with and entrain ambient water masses while descending to the abyss of the subpolar North Atlantic. I do not think there are other processes that can make the overflow waters significantly lighter. In general, the overflow waters are located too deep in the Labrador Sea, where the deepest convection occurs, to be accessed during convection in winter (Yashayaev, 2007). More importantly, for the overflow waters to be modified by convective mixing, the mixed layer would have to be sufficiently deep that it extends into the overflow layer. Since the ocean is stably stratified, the density of the mixed layer would then have to be at least the same as the density of the overflow layer. For this reason, deep convection would not make the overflow water lighter.

Line 642:

The deepest overflows are generally considered denser than $\sigma_\theta = 27.8$ kg/m$^3$ (Dickson and Brown, 1994).

Line 649:

This is a remarkably swift export of newly formed dense water, in particular considering that most of the transformation takes place within cyclonic gyres (e.g. Lavender *et al.*, 2000; Straneo *et al.*, 2003).

Line 656:

It is unclear to me how virtually all of the subpolar mode water is exported before undergoing further transformation to dense water, in particular considering that the residence time within the cyclonic gyres where most of the water mass transformation takes place may be on the order of years (Straneo *et al.*, 2003). Please elaborate.

Line 664:

Dense-water formation is not considered a "driver" of the AMOC (Kuhlbrodt *et al.*, 2007).

Lines 673, 681, and elsewhere:

Adding uncertainties to these estimates would be good.

Line 714:

Most high-latitude currents with substantial barotropic components closely follow density contours (e.g., Nøst and Isachsen, 2003). Instabilities in the West Greenland Current and formation of Irminger Rings may be a

more likely source of this signal (Fratantoni, 2001; Prater, 2002).

Line 727:
While deep convection at the boundary of the Labrador Sea may not have taken place in the 2000s, the boundary current system was ventilated during the more severe winters of the early- and mid-1990s (Pickart *et al.*, 1997).

Line 758:
I think it would be great to relate the overturning in depth space to the corresponding results obtained for the density space calculations. That would also integrate this section better within the rest of the manuscript.

Line 774:
This is the first proper discussion of the unresolved overflows. This is a major drawback of the coarse perimeter contour and likely has a substantial impact on the results. I think this discussion needs to be introduced much earlier in the manuscript.

Line 782:
More recent estimates of the Denmark Strait Overflow Water transport converge at values around 3.2-3.5 Sv (Harden *et al.*, 2016; Jochumsen *et al.*, 2017). This transport across the sill may then approximately double by entrainment as the dense water descends toward the abyss (Dickson and Brown, 1994). As such, VIKING20X may not be overestimating the overflow, although even in a relatively high-resolution model the overflows are probably not simulated very realistically.

Line 788:
More recent papers have made significant progress improving our understanding of the variability in Denmark Strait (Spall *et al.*, 2019; Lin *et al.*, 2020).

Line 792:
All of the reasons discussed in this paragraph may contribute, but the main cause of the poorly represented overflow water must be the low horizontal resolution along the perimeter contour.

Line 802:
Perhaps specify here that the dense water exiting the subpolar gyre in the North Atlantic Current continues to the north, into the Nordic Seas. As previously stated, this is an important result that demonstrates the importance of the subpolar gyre in preconditioning overturning in the Nordic Seas and thus modifies the conclusions of Petit *et al.* (2020).

Line 812:
The net sinking that occurs along the boundary (Spall and Pickart, 2000; Johnson *et al.*, 2019) may be another such process that is important to better understand, but difficult to address using this approach.

**Detailed comments:**

Lines 9, 105, 659, and elsewhere:
Oceans and Basins should be capitalized, also in plural.

Lines 16, 299, and 485:
Biscay is a province of Spain, I think Bay of Biscay would be more appropriate.

Line 28:
The acronym NAC should be defined at first usage.

Line 48:
"Arctic" by itself is an ill-defined term. Arctic Ocean would be better.

Lines 66, 134, 135, 138, 423, 460, 489, 492, and elsewhere:
I would have added at least one "the" to these lines.

Line 93:
"Deep mixing" is ambiguous. Do you mean convection/deep vertical mixing?

Line 149:
A search radius cannot be negative.

Line 198:
Is not an integral by definition cumulative?

Line 210:
Data are typically considered plural.

Line 223:
It should be: "...for an improve**d** representation..."

Line 225:
It should be: "...show **that** it realistically..."

Line 253:
It should be: "...surface Ekman **transports** capture..."

Line 257:
The Deep Western Boundary Current should be capitalized.

Line 275:
It should be: "...Using ERA5 month**ly** means..."

Line 290 and elsewhere
Scale-dependent is a compound modifier that should be hyphenated.

Line 316:
The comma should be removed.

Line 319:
The Labrador Current should be capitalized.

Lines 366 and 393:
Transport should not be capitalized.

Line 382:
The unit Sv is missing.

Line 463:
A comma is missing.

Line 592:
It should be: "...water **mass** transformation..."

S4:
Gulf Stream should be capitalized.

**References**

Chafik L, Rossby T. 2019. Volume, heat, and freshwater divergences in the Subpolar North Atlantic suggest the Nordic Seas as key to the state of the Meridional Overturning Circulation. *Geophysical Research Letters* **46**: doi:10.1029/2019GL082 110.

de Steur L, Pickart RS, Macrander A, Våge K, Harden B, Jónsson S, Østerhus S, Valdimarsson H. 2017. Liquid freshwater transport estimates from the East Greenland Current based on continuous measurements north of Denmark Strait. *Journal of Geophysical Research: Oceans* **122**: 93–109, doi:10.1002/2016JC012 106.

Dickson RR, Brown J. 1994. The production of North Atlantic Deep Water: Sources, rates and pathways. *Journal of Geophysical Research* **99**: 12 319–12 341, doi:10.1029/94JC00 530.

Fratantoni DM. 2001. North Atlantic surface circulation during the 1990's observed with satellite-tracked drifters. *Journal of Geophysical Research* **106**: 22 067–22 093.

Girton JB, Sanford TB. 2003. Descent and modification of the overflow plume in the Denmark Strait. *Journal of Physical Oceanography* **33**: 1351–1364.

Harden BE, Pickart RS, Valdimarsson H, Richards C, Våge K, de Steur L, Bahr F, Torres DJ, Børve E, Jónsson S, Macrander A, Østerhus S, Håvik L, Hattermann T. 2016. Upstream sources of the Denmark Strait Overflow: Observations from a high-resolution mooring array. *Deep Sea Research I* **112**: 94–112, doi:10.1016/j.dsr.2016.02.007.

Jochumsen K, Moritz M, Nunes N, Quadfasel D, Larsen KMH, Hansen B, Valdimarsson H, Jónsson S. 2017. Revised transport estimates of the Denmark Strait overflow. *Journal of Geophysical Research: Oceans* **122**: 3434–3450, doi:10.1002/2017JC012 803.

Johnson HL, Cessi P, Marshall DP, Schloesser F, Spall MA. 2019. Recent contributions of theory to our understanding of the Atlantic Meridional Overturning Circulation. *Journal of Geophysical Research: Oceans* **124**: doi:10.1029/2019JC015 330.

Jónsson S, Valdimarsson H. 2012. Water mass transport variability to the north Icelandic shelf, 1994-2010. *ICES Journal of Marine Science* : doi:10.1093/icesjms/fss024.

Kuhlbrodt T, Griesel A, Montoya M, Levermann A, Hofmann M, Rahmstorf S. 2007. On the driving processes of the Atlantic Meridional Overturning Circulation. *Reviews of Geophysics* **45**: RG2001, doi:10.1029/2004RG000 166.

Lavender KL, Davis RE, Owens WB. 2000. Mid-depth recirculation observed in the interior Labrador and Irminger Seas by direct velocity measurements. *Nature* **407**: 66–69.

Le Bras IAA, Straneo F, Holte J, de Jong MF, Holliday NP. 2020. Rapid export of waters formed by convection near the irminger sea's western boundary. *Geophysical Research Letters* **47**: doi:10.1029/2019GL085 989.

Lin P, Pickart RS, Jochumsen K, Moore GWK, Valdimarsson H, Fristedt T, Pratt LJ. 2020. "kinematic structure and dynamics of the denmark strait overflow from ship-based observations". *Journal of Physical Oceanography* **50**: 3235–3251, doi:10.1175/JPO–D–20–0095.1.

Lin P, Pickart RS, Torres DJ, Pacini A. 2018. "evolution of the freshwater coastal current at the southern tip of greenland". *Journal of Physical Oceanography* **48**: 2127–2140, doi:10.1175/JPO–D–18–0035.1.

Lozier MS, Li F, Bacon S, Bahr F, Bower AS, Cunningham SA, de Jong MF, de Steur L, de Young B, Fischer J, Gary SF, Greenan BJW, Holliday NP, Houk A, Houpert L, Inall ME, Johns WE, Johnson HL, Johnson C, Karstensen J, Koman G, Le Bras IA, Lin X, Mackay N, Marshall DP, Mercier H, Oltmanns M, Pickart RS, Ramsey AL, Rayner D, Straneo F, Thierry V, Torres DJ, Williams RG, Wilson C, Yang J, Yashayaev I, Zhao J. 2019. A sea change in our view of overturning in the subpolar North Atlantic. *Science* **363**: 516–521, doi:10.1126/science.aau6592.

Nøst OA, Isachsen PE. 2003. The large-scale time-mean ocean circulation in the Nordic Seas and Arctic Ocean estimated from simplified dynamics. *Journal of Marine Research* **61**: 175–210, doi:10.1357/002224003322005 069.

Østerhus S, Woodgate R, Valdimarsson H, Turrell WR, de Steur L, Quadfasel D, Olsen SM, Moritz M, Lee CM, Larsen KMH, Jónsson S, Johnson C, Jochumsen K, Hansen B, Curry B, Cunningham S, Berx B. 2019. Arctic Mediterranean exchanges: A consistent volume budget and trends in transports from two decades of observations. *Ocean Science* **15**: 379–399, doi:10.5194/os–15–379–2019.

Pacini A, Pickart RS. 2022. Meanders of the West Greenland Current near Cape Farewell. *Deep Sea Research I* **179**: doi:10.1016/j.dsr.2021.103 664.

Pacini A, Pickart RS, Bahr F, Torres DJ, Ramsey AL, Holte J, Karstensen J, Oltmanns M, Straneo F, Bras IAL, Moore GWK, de Jong MF. 2020. Mean conditions and seasonality of the West Greenland Boundary Current System near Cape Farewell. *Journal of Physical Oceanography* **50**: doi:10.1175/JPO–D–20–0086.1.

Petit T, Lozier MS, Josey SA, Cunningham SA. 2020. Atlantic deep water formation occurs primarily in the Iceland Basin and Irminger Sea by local buoyancy forcing. *Geophysical Research Letters* **47**: doi:10.1029/2020GL091 028.

Pickart RS, Spall MA. 2007. Impact of Labrador Sea convection on the North Atlantic Meridional Overturning Circulation. *Journal of Physical Oceanography* **37**: 2207–2227, doi:10.1175/JPO3178.1.

Pickart RS, Spall MA, Lazier JRN. 1997. Mid-depth ventilation in the western boundary current system of the sub-polar gyre. *Deep Sea Research I* **44**: 1025–1054.

Pickart RS, Torres DJ, Fratantoni PS. 2005. The East Greenland Spill Jet. *Journal of Physical Oceanography* **35**: 1037–1053.

Prater MD. 2002. Eddies in the labrador sea as observed by profiling rafos floats and remote sensing. *Journal of Physical Oceanography* **32**: 411–427, doi:10.1175/1520–0485(2002)032<0411:EITLSA>2.0.CO;2.

Semper S, Våge K, Pickart RS, Jónsson S, Valdimarsson H. 2022. Evolution and transformation of the North Icelandic Irminger Current along the north Iceland shelf. *Journal of Geophysical Research: Oceans* **127**: 10.1029/2021JC017 700.

Spall MA, Pickart RS. 2000. Where does dense water sink? A subpolar gyre example. *Journal of Physical Oceanography* **31**: 810–826.

Spall MA, Pickart RS, Lin P, von Appen W, Mastropole D, Valdimarsson H, Haine TWN, Almansi M. 2019. Frontogenesis and variability in Denmark Strait and its influence on overflow water. *Journal of Physical Oceanography* **49**: doi:10.1175/JPO–D–19–0053.1.

Straneo F, Pickart RS, Lavender KL. 2003. Spreading of Labrador Sea Water: An advective-diffusive study based on Lagrangian data. *Deep Sea Research I* **50**: 701–719.

Tsubouchi T, Våge K, Hansen B, Larsen KMH, Østerhus S, Johnson C, Jónsson S, Valdimarsson H. 2021. Increased ocean heat transport into the Nordic Seas and Arctic Ocean over the period 1993-2016. *Nature Climate Change* **11**: doi:10.1038/s41 558–020–00 941–3.

Yashayaev I. 2007. Hydrographic changes in the Labrador Sea, 1960-2005. *Progress in Oceanography* **73**: 242–276, doi:10.1016/j.pocean.2007.04.015.

Zantopp R, Fischer J, Visbeck M, Karstensen J. 2017. From interannual to decadal: 17 years of boundary current measurements at the exit of Labrador Sea. *Journal of Geophysical Research: Oceans* **122**: doi:10.1002/2016JC012 271.

---

## Author Comment (AC1)

**Responses to Reviewer 1**

This manuscript presents an innovative analysis of recent North Atlantic hydrographic measurements along with ancillary data sets to quantify the overturning in the subpolar gyre.

I think this is a well-written and very interesting paper that significantly contributes to the understanding of water mass transformation in the subpolar North Atlantic. In my opinion, the analysis has two major weaknesses, namely (i) that the perimeter contour for large parts of the domain is oriented along the major boundary current system such that the cross-contour component of the flow is a small residual relative to the along-contour flow and (ii) that the perimeter contour has too coarse resolution to capture important components of the circulation, in particular the overflows through Denmark Strait and Faroe Bank Channel. I am not sure that these two weaknesses can be robustly addressed, but I think that at least a more extensive discussion of these two concerns is necessary. I also have a few other comments that I hope the authors will consider. Finally, I would like to emphasize that I think this manuscript ingeniously utilizes existing observations to address an important and challenging scientific question; I support the effort and encourage the authors to submit a revised version of the manuscript.

General comments:

Most of the major currents of the subpolar gyre boundary current system have substantial flow along the 1000 m isobath, such as the East Greenland (Le Bras et al., 2020) and West Greenland Currents (Pacini et al., 2020), and the dominant isopycnal slope along this depth contour is across the boundary current system. These currents are to varying extent subject to meanders and instabilities (e.g., Prater, 2002; Pacini and Pickart, 2022), which introduce substantial variability in the measurements. Quantifying the cross-slope geostrophic flow from the isopycnal slope along a depth contour that is characterized by substantial and vigorous dynamics is not optimal. Choosing a deeper isobath for the perimeter contour would likely alleviate the problem. While there are other good reasons for choosing the 1000 m depth contour, I think more robust estimates of cross-contour flow would be obtained along a deeper isobath. I will not advocate that the analysis is redone, but would like to see this issue more extensively discussed in the paper.

For the horizontal gridding along the perimeter contour, a resolution of 150 km was used. The coarse resolution may suffice for the large-scale, geostrophic interior-boundary exchange. However, very important contributions to this exchange occurs on much smaller spatial scales, in particular eddies and deep overflows from the Nordic Seas, but also currents such as the Deep Western Boundary Current are not resolved at this scale. While eddies may roughly balance in- and outward fluxes across the perimeter contour, the overflows and the water masses they entrain are crucial inflows into the subpolar gyre that will not have been properly accounted for. Downstream of Denmark Strait, the overflow plume has a spatial scale of much less than 100 km and rapidly descends beneath the 1000 m depth contour (Dickson and Brown, 1994; Girton and Sanford, 2003) – hence the Denmark Strait overflow cannot be the main source of inflow near Cape Farewell.

General response:

Thank-you for the broad and constructive review. We largely agree with the suggested changes and will implement where indicated in our responses.

Reviewer 1 highlighted two key concerns with the method. The first concern related to the choice of the 1000 m contour as the domain boundary, as in some regions the intense boundary currents

bisect this contour. As correctly highlighted, the cross-contour component of the flow used in this study is small compared to the along contour component.

The choice of the 1000 m isobath was motivated by several considerations. First, Argo profiles play a major role in the observational analysis, so if the curtain of data is to be in contact with the seabed, we are limited to isobaths shallower than 2000 m. Second, the choice of isobaths greater than ~1500 m result in Rockall Plateau being excluded from the domain. One of the main regions where the boundary currents are offshore of the 1000 m contour is off southwest Greenland (the WGC). In this region the continental slope is very steep and a choice of 2000 m or more as the reference isobath does not prevent the boundary current crossing the contour. In fact, when recreating the analysis in VIKING20X using the 2400 m isobath as the reference contour, we found the cross-boundary flows became more intense. This appears to be because the WGC crosses the 2400 m isobath more abruptly than it crosses the 1000 m isobath. The intense boundary currents also remain offshore of the 2400 m isobath along the western Labrador Sea.

To exclude all the major boundary currents from the domain, we might consider offsetting the boundary contour a set distance offshore from an isobath. However, the convenient geostrophic constraint provided by a constant-depth curtain of data is then lost, and we exacerbate the problem of undiagnosed flow under the data curtain. As suggested by Reviewer 1, we have expanded the discussion of the choice of contour to include some of the points raised here.

The second concern was with our choice of resolution for the climatology, and whether this resulted in the exclusion of small-scale but dynamically important features such as overflows. We did try different horizontal resolutions, and we thought the 150km horizontal grid size the best compromise between spatial resolution and the available profile density. In particular, we wanted a climatology which could be robustly split into 4 seasons without regions of poor coverage emerging. While the 150 km resolution was a good compromise for the boundary as a whole, Reviewer 1 highlights that some important small-scale features such as the overflows may be lost due to resolution and smooth scale.

To investigate the extent to which the resolution impacted our ability to resolve the overflows, we examined the raw Argo and CTD profiles in the dataset for evidence of the TS characteristics of overflow water at the expected locations of the Denmark Strait Overflow and Faroe Bank Channel Overflow. We found that very few (<10) profiles featured the expected TS properties. One reason for this finding appears to be the geometry of the 1000 m data cut-off relative to the seabed; the near-bed overflows only intersect with the data collection region close where it is in contact with the bed. Increasing the resolution would therefore not substantially increase the prominence of the overflows as they are not being captured in the raw profiles. See the geometry of the data collection region relative to the seabed in Fig. 12 in the manuscript. Note also that choosing a deeper reference contour would not alleviate this problem. We list several other factors which might limit our ability to properly sample the overflows using scattered CTD and Argo profiles in the manuscript.

Further, we argue that even if the overflows were perfectly resolved, a substantial portion of their flow is ageostrophic (as evidenced by the VIKING20X analysis) and so would not contribute to the volume transport estimates. We already stress that the omission of the overflows will have little impact on the overturning findings, as the overturning streamfunction is integrated from the surface downwards, and the overflows are too dense to undergo further transformation in the domain. However, we will expand the discussion of the potential impact on the heat and freshwater estimates in the revised manuscript and introduce the concept earlier in the manuscript.

Specific comments:

Line 25:

Is the minimum overturning in fall mainly comprised of the overflows from the Nordic Seas, which would form a steady baseline with minimal seasonal variability, while the maximum in spring also includes dense water formed within the subpolar gyre? Or is the surface Ekman component also an important source of variability? I think it would be good to specify the cause of this seasonal variability already in the abstract.

Surface Ekman appears to be the main driver of this seasonality. Added a sentence to clarify.

Line 36: Including the estimate of 30526 TW across the Greenland-Scotland Ridge by Tsubouchi et al. (2021) would be another very relevant point of comparison here.

Thank-you for the suggestion. Added to text.

Line 48: Another important component of the return flow from the Arctic Ocean and Nordic Seas is the surface outflow of Polar Water in the East Greenland Current (de Steur et al., 2017).

Good point, added to text.

Line 60: The relatively low impact of water mass transformation in the Labrador Sea on the AMOC was known also prior to OSNAP (e.g., Pickart and Spall, 2007).

Modified text to include this point.

Line 64: The importance of water mass transformation north of the Greenland-Scotland Ridge for supply of dense water to the lower limb of the AMOC should not be underestimated (e.g., Chafik and Rossby, 2019).

Added a note clarifying this point.

Line 99: Another important dynamical consideration is that, apart from the overflows, most of the sinking occurs along the boundary (Spall and Pickart, 2000; Johnson et al., 2019)

Modified text to include this point.

Line 148: If the minimal search radius is 150 km, equal to the distance between grid points, most of the profiles are probably used in more than one grid cell. Did you apply any weighting to emphasize the contributions of profiles closer to the grid point or the perimeter contour?

We did not attempt any weighting in the along-contour direction, or in the distance from the perimeter contour. Added a note in the text to clarify this point. As property gradients are high in the across-contour direction, weighting by distance from the contour may result in unpredictable responses to data scatter. As noted in the next point, we feel that the uncertainty analysis provides a robust view of the errors which might result from the existing gridding approach.

Line 199: This is a great approach to estimate the statistical uncertainty inherent in the data set. Providing the measurement errors that are also inherent in the data set, such that the magnitudes of the statistical and measurement uncertainties can directly be compared, would also be good.

We now include the measurement errors associated with CTDs, Argo and satellite ADT. We found that the scatter in results which might be expected due to measurement error was negligible when compared with the statistical uncertainty.

Line 210: Please provide some more details regarding the calculation of the surface Ekman transport. For example, what have you taken to be the depth of the Ekman layer?

For the flux and overturning calculations, the Ekman transports are added to velocities in the top 20 m cell. They therefore act on the corresponding top cells of the gridded temperature and salinity.

Line 258: Important flows on relatively small scale such as the overflows from the Nordic Seas and the East Greenland Spill Jet (Pickart et al., 2005) are not properly resolved. Perhaps EN4 at 47°N is not the sole cause of the imbalance, if these features, along with the Deep Western Boundary Current, substantially contribute to the imbalance?

As discussed later in the manuscript, we found a good qualitative agreement between the calculated geostrophic velocities and those diagnosed in VIKING20X, suggesting that the observations were capturing the important flows across the boundary above 1000 m.  The 'remainder' term in the model (including most of the overflow transports) is flow that we would not be able to include using the geostrophic approach, even with perfect sampling.

Line 283: What are the length scales over which the satellite absolute dynamic topography product was smoothed? On line 183 it is stated that smoothing was applied to mimic the smoothing inherent in the hydrographic gridding process. Is the resulting length scale of the eddy kinetic energy consistent with eddies scaled by the Rossby radius, or were all eddies, to the extent that they were represented in the raw satellite record, removed by the smoothing procedure?

These instances reflect different treatments of the satellite ADT product for different purposes.  There is no smoothing applied to the ADT prior to computing the eddy kinetic energy or diffusive fluxes.  Added text to clarify this point.

Line 308: Offshore fluxes of freshwater from the Greenland shelf into the interior Labrador Sea near Cape Farewell (Lin et al., 2018) and farther north where the West Greenland Current encounters steep topography and becomes unstable (e.g., Fratantoni, 2001; Prater, 2002) are likely major contributors to the cold, fresh low-density layer. Both of these processes are primarily eddy-driven, hence postulating that a portion of the West Greenland Current crosses into the interior supbolar gyre may not be necessary. These processes will not be resolved at 150 km horizontal resolution. Given the turbulent nature of these fluxes, it is also not obvious that there will be a consistent geostrophic flux across the perimeter contour.

This is a good point, added text to clarify the probable cause of the cold, fresh intrusion.  As you say, these fluxes are primarily turbulent and may not be associated with a positive geostrophic flow so we still state that this may be due to the WGC moving into deeper water in this region.

Lines 349 and 491: Most of the Atlantic Water inflow from the subpolar gyre to the Nordic Seas takes place east of Iceland, roughly evenly split on either side of the Faroe Islands (Østerhus et al., 2019). Given the course resolution of the perimeter contour, I think it is more appropriate to ascribe this flow to the Iceland-Scotland Ridge rather than the Wyville Thomson Ridge (note that Thomson is spelled without a p).

Agreed; updated text.

Line 351: Note that there is also some flow of Atlantic Water northward through Denmark Strait (Jonsson and Valdimarsson, 2012; Semper et al., 2022).

Thank-you; added to text.

Line 352: What is the magnitude of the retroflection of the East Greenland Current near Cape Farewell?

5.1 Sv flow from the EGC into the central Irminger basin (Holliday et al., 2007); added to text.

Line 357: An export of 12 Sv from the subpolar gyre to the Labrador shelf is immense. Is this a realistic number? Could this be related to water sinking along the boundary (e.g., Johnson et al., 2019) or merge with the boundary current system (a substantial portion of which appears to be inshore of the 1000 m isobath, Zantopp et al., 2017)?

The description stating that ~12 Sv flowed onto the shelf was misleading.  Inference of on-shelf flow in fact stops at ~9200 km, at which point the flow is not onto the shelf but through and over the Flemish Cap.  Whilst the core of the boundary current is inshore of the 1000 m isobath, at 53 N it extends 75-100 km offshore of the 1000 m isobath (e.g. Fig. 8 in Zantopp et al., 2017).  If we assume this region averages 10 cm s$^{-1}$ (referring again to Zantopp et al., 2017) this suggests several Sv of the boundary current could be within the boundary contour (and above 1000 m) at this latitude. The remainder is gained just south of the OSNAP crossing.  The volume flowing south through the Flemish Pass could account for 6-10 Sv (Petrie and Buckley 1996).  Added the above transport estimates to the manuscript.

Line 450: Good discussion of missing contributions. While the model has a much higher resolution, how confident can you be that it is able to realistically capture these features?

These small-scale baroclinic features are always going to be challenging for a basin-scale model to accurately represent, but their contributions to the volume budget of the SPG do appear to be reasonably consistent with observation campaigns (e.g. Biastoch et al., 2021, Harden et al., 2016; Jochumsen et al., 2017).

Lines 493, 631, and 696: It is not obvious why there should be a lower layer inflow in the vicinity of Cape Farewell. This is too far south of Denmark Strait to be ascribed to the overflow (Dickson and Brown, 1994; Girton and Sanford, 2003), but perhaps the East Greenland Spill Jet (Pickart et al., 2005) contributes? Please elaborate.

We should probably reiterate that in the context of the Irminger and Labrador Basins, our definition of the lower layer (> 27.54 kg/m3) is still quite light.  Examining Figs. 3c and d, it's clear that this density class accounts for all transport below ~150 dbar.  Given the water properties in this region are those of the EGC (Holliday et al., 2007), it seems likely that this inflow is at least in part due to the ~5.1 Sv retroflection of the EGC into the Irminger Sea observed by Holiday et al. (2007).  As we note in the manuscript, another factor could be the tendency of the EGC to track deeper isobaths west of Cape Farewell as the shelf edge steepens.  Added notes to this effect at the suggested locations in the manuscript.

Line 525: This estimate of advective flux across the Iceland-Scotland Ridge appears to be in reasonably good agreement with the estimate of Tsubouchi et al. (2021).

Thank-you for pointing this out, noted in text.

Lines 572 and 726: There is substantial discussion of the high EKE west of Greenland in the literature (e.g., Fratantoni, 2001; Prater, 2002).

Thank-you; added to text.

Line 599: The overturning across the Greenland-Scotland Ridge would be another vital point of comparison (e.g., Østerhus et al., 2019; Tsubouchi et al., 2021).

Added to text.

Line 618: How does the density surface of 27.30 kg/m3 for maximum overturning compare to similar results from Lozier et al. (2019) and Petit et al. (2020)?

27.30 kg/m3 is substantially lighter than the density of maximum overturning found by Lozier et al. (2019) (27.66 kg/m3) and Petit et al. (2020) (27.55 kg/m3).  These comparisons added to the text.

Line 623: This is an important result, which substantially modifies the conclusions of Petit et al. (2020). Without velocity measurements, they considered this water mass transformation part of the overturning in the subpolar gyre, and concluded that more deep-water formation occurs in the subpolar gyre than in the Nordic Seas. You have demonstrated that a substantial portion of this

intermediate-density water continues to the north, into the Nordic Seas, where it is further transformed. As such, densification in the subpolar gyre preconditions further water mass transformation in the Nordic Seas and is thereby important for the North Atlantic overturning, but it is not appropriate to ascribe that part of the water mass transformation to overturning in the subpolar gyre, since the water proceeds into the Nordic Seas in the upper layer rather than returning to the south at depth.

Thank-you for highlighting this point. We have modified the text to clarify the role of pre-conditioning based on our findings, and how this differs from the findings of Petit et al. (2020).

Line 639: Overflow waters from the Nordic Seas become lighter as they mix with and entrain ambient water masses while descending to the abyss of the subpolar North Atlantic. I do not think there are other processes that can make the overflow waters significantly lighter. In general, the overflow waters are located too deep in the Labrador Sea, where the deepest convection occurs, to be accessed during convection in winter (Yashayaev, 2007). More importantly, for the overflow waters to be modified by convective mixing, the mixed layer would have to be sufficiently deep that it extends into the overflow layer. Since the ocean is stably stratified, the density of the mixed layer would then have to be at least the same as the density of the overflow layer. For this reason, deep convection would not make the overflow water lighter.

Thank-you for pointing this out. Modified text to state that this is most likely due to mixing and entrainment.

Line 642: The deepest overflows are generally considered denser than sq = 27:8 kg/m3 (Dickson and Brown, 1994).

Modified threshold and added reference to text.

Line 649: This is a remarkably swift export of newly formed dense water, in particular considering that most of the transformation takes place within cyclonic gyres (e.g. Lavender et al., 2000; Straneo et al., 2003).

Yes, it seems a more likely cause of the springtime overturning maximum is that winter surface Ekman forcing acts to suppress overturning, shifting the peak to the spring, in a similar manner to that seen in OSNAP (Li et al., 2021a; Petit et al., 2020; Petit et al., 2021). If we remove surface Ekman forcing from the volume budget, the overturning peak occurs in winter instead. Changed text to reflect this.

Line 656: It is unclear to me how virtually all of the subpolar mode water is exported before undergoing further transformation to dense water, in particular considering that the residence time within the cyclonic gyres where most of the water mass transformation takes place may be on the order of years (Straneo et al., 2003). Please elaborate.

Only half the SPMW is exported. This sentence was confusing and has been rephrased.

Line 664: Dense-water formation is not considered a "driver" of the AMOC (Kuhlbrodt et al., 2007).

Changed to "important source of dense water masses for the lower limb of the AMOC".

Lines 673, 681, and elsewhere: Adding uncertainties to these estimates would be good.

Added error estimates where suggested, and at other appropriate locations.

Line 714: Most high-latitude currents with substantial barotropic components closely follow density contours (e.g., Nøst and Isachsen, 2003). Instabilities in the West Greenland Current and formation of Irminger Rings may be a more likely source of this signal (Fratantoni, 2001; Prater, 2002).

Agreed, changed text and added references to reflect this.

Line 727: While deep convection at the boundary of the Labrador Sea may not have taken place in the 2000s, the boundary current system was ventilated during the more severe winters of the early- and mid-1990s (Pickart et al., 1997).

Thank-you for highlighting this.  Added to text.

Line 758: I think it would be great to relate the overturning in depth space to the corresponding results obtained for the density space calculations. That would also integrate this section better within the rest of the manuscript.

The depth of density contours varies widely around the SPG so it is hard to generalise the overturning in depth space for the entire SPG boundary.  We have enhanced the visibility of key overturning isopycnals in Fig. 3d (geostrophic velocity) and added them to Fig. 3c (density).  We have also added a label to the overturning stream function for the 47N transect (Fig. 9c) to indicate that the 27.7 contour is at approximately 1000 m.

Line 774: This is the first proper discussion of the unresolved overflows. This is a major drawback of the coarse perimeter contour and likely has a substantial impact on the results. I think this discussion needs to be introduced much earlier in the manuscript.

As previously discussed, the inability to resolve the overflows is primarily a sampling problem and not a resolution problem.  We have increased signposting to this section throughout the manuscript and have also noted the absence of the expected overflows in the Hydrography section of the results (Section 3.1).  There is also an introduction to the missing overflows and other ageostrophic processes in the VIKING20XS analysis (Section 3.3).

Line 782: More recent estimates of the Denmark Strait Overflow Water transport converge at values around 3.2- 3.5 Sv (Harden et al., 2016; Jochumsen et al., 2017). This transport across the sill may then approximately double by entrainment as the dense water descends toward the abyss (Dickson and Brown, 1994). As such, VIKING20X may not be overestimating the overflow, although even in a relatively high-resolution model the overflows are probably not simulated very realistically.

Good point; added to text.

Line 788: More recent papers have made significant progress improving our understanding of the variability in Denmark Strait (Spall et al., 2019; Lin et al., 2020).

Thank-you; added references.

Line 792: All of the reasons discussed in this paragraph may contribute, but the main cause of the poorly represented overflow water must be the low horizontal resolution along the perimeter contour.

Whilst we partially agree, even In the raw CTD data, relatively few profiles capture true DSO water.  Presumably unless the profile is very close to the boundary contour the overflow passes below the 1000 m curtain in our analysis.  We could increase the proportion of 'overflow' profiles by reducing the offshore search area, but this would have a detrimental effect on the data density.  But we have added a note in the text to acknowledge this point.

Line 802: Perhaps specify here that the dense water exiting the subpolar gyre in the North Atlantic Current continues to the north, into the Nordic Seas. As previously stated, this is an important result that demonstrates the importance of the subpolar gyre in preconditioning overturning in the Nordic Seas and thus modifies the conclusions of Petit et al. (2020).

Thank-you, modified text to clarify.

Line 812: The net sinking that occurs along the boundary (Spall and Pickart, 2000; Johnson et al., 2019) may be another such process that is important to better understand, but difficult to address using this approach.

Good point; added to text.

**Detailed comments:**

Lines 9, 105, 659, and elsewhere: Oceans and Basins should be capitalized, also in plural.

Comment addressed.

Lines 16, 299, and 485: Biscay is a province of Spain, I think Bay of Biscay would be more appropriate.

Comment addressed.

Line 28: The acronym NAC should be defined at first usage.

Comment addressed.

Line 48: "Arctic" by itself is an ill-defined term. Arctic Ocean would be better.

Comment addressed.

Lines 66, 134, 135, 138, 423, 460, 489, 492, and elsewhere: I would have added at least one "the" to these lines.

Thank-you for highlighting these omissions.

Line 93: "Deep mixing" is ambiguous. Do you mean convection=deep vertical mixing?

Yes, the intended meaning was convection. Text amended.

Line 149: A search radius cannot be negative.

Agreed. Text amended.

Line 198: Is not an integral by definition cumulative?

Agreed. Text amended.

Line 210: Data are typically considered plural.

Text amended.

Line 223: It should be: "...for an improved representation..."

Comment addressed.

Line 225: It should be: "...show that it realistically..."

Text amended.

Line 253: It should be: "...surface Ekman transports capture..."

Text amended.

Line 257: The Deep Western Boundary Current should be capitalized.

Text amended.

Line 275: It should be: "...Using ERA5 monthly means..."

Text amended.

Line 290 and elsewhere Scale-dependent is a compound modifier that should be hyphenated.

Text amended.

Line 316: The comma should be removed.

Text amended.

Line 319: The Labrador Current should be capitalized.

Text amended.

Lines 366 and 393: Transport should not be capitalized.

Text amended.

Line 382: The unit Sv is missing.

Text amended.

Line 463: A comma is missing.

Text amended.

Line 592: It should be: "...water mass transformation..."

Text amended.

S4: Gulf Stream should be capitalized.

Text amended.

References

Chafik L, Rossby T. 2019. Volume, heat, and freshwater divergences in the Subpolar North Atlantic suggest the

Nordic Seas as key to the state of the Meridional Overturning Circulation. Geophysical Research Letters 46:

doi:10.1029/2019GL082 110.

de Steur L, Pickart RS, Macrander A, V°age K, Harden B, J´onsson S, Østerhus S, Valdimarsson H. 2017. Liquid

freshwater transport estimates from the East Greenland Current based on continuous measurements north of

Denmark Strait. Journal of Geophysical Research: Oceans 122: 93–109, doi:10.1002/2016JC012 106.

Dickson RR, Brown J. 1994. The production of North Atlantic Deep Water: Sources, rates and pathways.

Journal of Geophysical Research 99: 12 319–12 341, doi:10.1029/94JC00 530.

Fratantoni DM. 2001. North Atlantic surface circulation during the 1990's observed with satellite-tracked

drifters. Journal of Geophysical Research 106: 22 067–22 093.

Girton JB, Sanford TB. 2003. Descent and modification of the overflow plume in the Denmark Strait. Journal

of Physical Oceanography 33: 1351–1364.

Harden BE, Pickart RS, Valdimarsson H, Richards C, V°age K, de Steur L, Bahr F, Torres DJ, Børve E,

J´onsson S, Macrander A, Østerhus S, H°avik L, Hattermann T. 2016. Upstream sources of the Denmark

Strait Overflow: Observations from a high-resolution mooring array. Deep Sea Research I 112: 94–112,

doi:10.1016/j.dsr.2016.02.007.

Jochumsen K, Moritz M, Nunes N, Quadfasel D, Larsen KMH, Hansen B, Valdimarsson H, J´onsson S. 2017.

Revised transport estimates of the Denmark Strait overflow. Journal of Geophysical Research: Oceans 122:

3434–3450, doi:10.1002/2017JC012 803.

Johnson HL, Cessi P, Marshall DP, Schloesser F, Spall MA. 2019. Recent contributions of theory to our

understanding of the Atlantic Meridional Overturning Circulation. Journal of Geophysical Research: Oceans

124: doi:10.1029/2019JC015 330.

J´onsson S, Valdimarsson H. 2012. Water mass transport variability to the north Icelandic shelf, 1994-2010.

ICES Journal of Marine Science : doi:10.1093/icesjms/fss024.

Kuhlbrodt T, Griesel A, Montoya M, Levermann A, Hofmann M, Rahmstorf S. 2007. On the driving

processes of the Atlantic Meridional Overturning Circulation. Reviews of Geophysics 45: RG2001,

doi:10.1029/2004RG000 166.

Lavender KL, Davis RE, Owens WB. 2000. Mid-depth recirculation observed in the interior Labrador and

Irminger Seas by direct velocity measurements. Nature 407: 66–69.

Le Bras IAA, Straneo F, Holte J, de Jong MF, Holliday NP. 2020. Rapid export of waters formed by convection

near the irminger sea's western boundary. Geophysical Research Letters 47: doi:10.1029/2019GL085 989.

Lin P, Pickart RS, Jochumsen K, Moore GWK, Valdimarsson H, Fristedt T, Pratt LJ. 2020. "kinematic

structure and dynamics of the denmark strait overflow from ship-based observations". Journal of Physical

Oceanography 50: 3235–3251, doi:10.1175/JPO–D–20–0095.1.

Lin P, Pickart RS, Torres DJ, Pacini A. 2018. "evolution of the freshwater coastal current at the southern tip of

greenland". Journal of Physical Oceanography 48: 2127–2140, doi:10.1175/JPO–D–18–0035.1.

Lozier MS, Li F, Bacon S, Bahr F, Bower AS, Cunningham SA, de Jong MF, de Steur L, de Young B, Fischer

J, Gary SF, Greenan BJW, Holliday NP, Houk A, Houpert L, Inall ME, Johns WE, Johnson HL, Johnson

C, Karstensen J, Koman G, Le Bras IA, Lin X, Mackay N, Marshall DP, Mercier H, Oltmanns M, Pickart

RS, Ramsey AL, Rayner D, Straneo F, Thierry V, Torres DJ, Williams RG, Wilson C, Yang J, Yashayaev I,

Zhao J. 2019. A sea change in our view of overturning in the subpolar North Atlantic. Science 363: 516–521,

doi:10.1126/science.aau6592.

Nøst OA, Isachsen PE. 2003. The large-scale time-mean ocean circulation in the Nordic Seas and

Arctic Ocean estimated from simplified dynamics. Journal of Marine Research 61: 175–210,

doi:10.1357/002224003322005 069.

Østerhus S, Woodgate R, Valdimarsson H, Turrell WR, de Steur L, Quadfasel D, Olsen SM, Moritz M, Lee

CM, Larsen KMH, J´onsson S, Johnson C, Jochumsen K, Hansen B, Curry B, Cunningham S, Berx B. 2019.

Arctic Mediterranean exchanges: A consistent volume budget and trends in transports from two decades of

observations. Ocean Science 15: 379–399, doi:10.5194/os–15–379–2019.

Pacini A, Pickart RS. 2022. Meanders of the West Greenland Current near Cape Farewell. Deep Sea Research

I 179: doi:10.1016/j.dsr.2021.103 664.

Pacini A, Pickart RS, Bahr F, Torres DJ, Ramsey AL, Holte J, Karstensen J, Oltmanns M, Straneo F, Bras IAL,

Moore GWK, de Jong MF. 2020. Mean conditions and seasonality of the West Greenland Boundary Current

System near Cape Farewell. Journal of Physical Oceanography 50: doi:10.1175/JPO–D–20–0086.1.

Petit T, Lozier MS, Josey SA, Cunningham SA. 2020. Atlantic deep water formation occurs primarily

in the Iceland Basin and Irminger Sea by local buoyancy forcing. Geophysical Research Letters 47:

doi:10.1029/2020GL091 028.

Pickart RS, Spall MA. 2007. Impact of Labrador Sea convection on the North Atlantic Meridional Overturning

Circulation. Journal of Physical Oceanography 37: 2207–2227, doi:10.1175/JPO3178.1.

Pickart RS, Spall MA, Lazier JRN. 1997. Mid-depth ventilation in the western boundary current system of the

sub-polar gyre. Deep Sea Research I 44: 1025–1054.

Pickart RS, Torres DJ, Fratantoni PS. 2005. The East Greenland Spill Jet. Journal of Physical Oceanography

35: 1037–1053.

Prater MD. 2002. Eddies in the labrador sea as observed by profiling rafos floats and remote sensing. Journal

of Physical Oceanography 32: 411–427, doi:10.1175/1520–0485(2002)032<0411:EITLSA>2.0.CO;2.

Semper S, V°age K, Pickart RS, J´onsson S, Valdimarsson H. 2022. Evolution and transformation of the North

Icelandic Irminger Current along the north Iceland shelf. Journal of Geophysical Research: Oceans 127:

10.1029/2021JC017 700.

Spall MA, Pickart RS. 2000. Where does dense water sink? A subpolar gyre example. Journal of Physical

Oceanography 31: 810–826.

Spall MA, Pickart RS, Lin P, von Appen W, Mastropole D, Valdimarsson H, Haine TWN, Almansi M. 2019.

Frontogenesis and variability in Denmark Strait and its influence on overflow water. Journal of Physical

Oceanography 49: doi:10.1175/JPO–D–19–0053.1.

Straneo F, Pickart RS, Lavender KL. 2003. Spreading of Labrador Sea Water: An advective-diffusive study

based on Lagrangian data. Deep Sea Research I 50: 701–719.

Tsubouchi T, V°age K, Hansen B, Larsen KMH, Østerhus S, Johnson C, J´onsson S, Valdimarsson H. 2021.

Increased ocean heat transport into the Nordic Seas and Arctic Ocean over the period 1993-2016. Nature

Climate Change 11: doi:10.1038/s41 558–020–00 941–3.

Yashayaev I. 2007. Hydrographic changes in the Labrador Sea, 1960-2005. Progress in Oceanography 73:

242–276, doi:10.1016/j.pocean.2007.04.015.

Zantopp R, Fischer J, Visbeck M, Karstensen J. 2017. From interannual to decadal: 17 years of boundary

current measurements at the exit of Labrador Sea. Journal of Geophysical Research: Oceans 122:

doi:10.1002/2016JC012 271.

**References**

Biastoch, A., Schwarzkopf, F.U., Getzlaff, K., Rühs, S., Martin, T., Scheinert, M., Schulzki, T., Handmann, P.,

Hummels, R. and Böning, C.W.: Regional imprints of changes in the Atlantic meridional overturning circulation in the eddy-rich ocean model VIKING20X, Ocean Sci. Discuss., 1-52, 2021.

Harden, B.E., Pickart, R.S., Valdimarsson, H., Våge, K., de Steur, L., Richards, C., Bahr, F., Torres, D., Børve, E., Jónsson, S. and Macrander, A.: Upstream sources of the Denmark Strait Overflow: Observations from a high-resolution mooring array, Deep-Sea Res. Pt. I, 112, 94-112, doi:10.1016/j.dsr.2016.02.007, 2016.

Holliday, N. P., Meyer, A., Bacon, S., Alderson, S. G. and de Cuevas, B.: Retroflection of part of the east Greenland current at Cape Farewell, Geophys. Res. Lett., 34(7), 7609, doi:10.1029/2006GL029085, 2007

Jochumsen, K., Moritz, M., Nunes, N., Quadfasel, D., Larsen, K.M., Hansen, B., Valdimarsson, H. and Jonsson, S.: Revised transport estimates of the Denmark S trait overflow, J. Geophys. Res. Ocean., 122(4), 3434-3450, doi:10.1002/2017JC012803, 2017.

Petrie, B. and Buckley, J.: Volume and freshwater transport of the Labrador Current in Flemish Pass. J. Geophys. Res. Ocean., 101(C12), 28335-28342, doi:10.1029/96JC02779, 1996.

Zantopp, R., Fischer, J., Visbeck, M. and Karstensen, J.: From interannual to decadal: 17 years of boundary current transports at the exit of the Labrador Sea, J. Geophys. Res. Ocean., 122(3), 1724-1748, doi: 10.1002/2016JC012271, 2017.

---

## Author Comment (AC2)

**Responses to Reviewer 2**

This paper creates a novel climatology of the subpolar North Atlantic around the 1000 m isobath and across 47N and discusses properties and fluxes across and with this region. The techniques used are interesting and the climatology looks great. I have reservations about the use of EN4 at 47 N that I think could be investigated further. I think the discussion and observations are good and interesting.

My major comment on the paper is that it could be much more focused. The introduction covers much more material than the results address. The question that the climatology and calculations are addressing could be framed much more succinctly. Likewise in the discussion and conclusions, there needs to be a closing of the loop back. E.g. the discussion around Fig. 12 was very interesting but I wasn't sure what question this was addressing.

I have a long list but these are minor comments, the only major comment is a tightening up of the framing of the results.

General response:

Thank-you for the review.  In response to the main concerns of Reviewer 2, we have shortened and restructured the abstract and introduction, improved the framing of the key findings and increased the signposting to the main points raised in the discussion.  Some of these changes also incorporate suggestions by Reviewer 1.  We largely agree with the minor comments and will implement the suggested changes where indicated.

Minor comments:

Why were Argo velocities not used? The dataset seems dominated by Argo Fig 2b

For the main investigation (across-boundary transports) we investigated the ANDRO dataset but found that the interpolation scheme was unsuitable for the boundary.  The relatively high along-slope velocities also meant that any errors in inferred velocity from Argo trajectories would be magnified when the small across-slope component was investigated.  Similarly, for the estimate of bottom Ekman flow we found that ANDRO was not suitable for the analysis of boundary currents due to the proximity to the continental slope.

I think the abstract is too long and could be shortened to 2 paragraphs. Too much intro material in paragraph 3 of the abstract especially.

We will shift the focus of the abstract towards the key points and shorten in the revised manuscript.

L32. This definition of the AMOC is not correct: the AMOC (uniquely) transports heat across the tropics from the South Atlantic

Agreed, modified text to clarify.

L53, no need to complicate with the drifter results

Agreed, this point has been removed from the restructured introduction.

L58, canonical -> generally accepted

Text amended.

L64, to the mean what? This line throws the paragraph out. If you're considering processes north of GSR, then your first sentence should consider these also i.e. GSR overflows + entrainment in addition to Lab Sea processes are fundamental to AMOC functioning.

Text amended also in response to a similar comment by Reviewer 1.

L67, 'they' is ambiguous here. I presume you mean Lab Sea density anomalies?

Text amended.

L70, don't see why you're bringing in subpolar mode water

Restructured paragraph for clarity.

L78, add 'in the eastern basin'

Text modified.

The introduction is very general. It should be more focused to frame this study rather than a general subpolar gyre introduction.

As discussed in the general response, we have shortened and restructured the introduction to better frame the research questions.

Fig. 2. > Radon transform for analysis of propagation speeds in Fig. 2.

As for the ANDRO product, the proximity of the boundary makes the analysis of float propagation speeds problematic. For the bottom Ekman order of magnitude discussion we only require a crude estimate, and the radon transform approach would only serve to narrow the uncertainties around the supplied figure.

> Not much data prior to 2008.

Argo data is sparser before 2008, but the regular CTD transects which bisect the dataset provide reasonable continued coverage. As Argo was still being populated in the 2000s, there will inevitably be some temporal bias towards later years but we feel that the early 2000s still makes a valuable contribution to the climatology.

> Higher propagation speeds upstream of FSC.

We don't see much evidence of higher propagation speeds upstream of the FSC. Floats in the Rockall Trough are rarely entrained in the European slope current as it is too narrow and shallow for a typical Argo drift profile.

Propagation speeds are only relevant for the Argo data, not the CTD data (unless you're telling us about the speed of the ship). Can ship CTD data be removed from Fig. 2b.

We have now coloured the ship CTD data distinguish it from the Argo profiles in Fig. 2a and b.

L144. What is the justification for using a much longer search radius in the along bathymetry direction than cross bathymetry, limited to 75 km?

Cross-bathymetry property gradients are much greater than along-bathymetry gradients. As we were treating the dataset as a nominal transect along the 1000 m isobath (to enable geostrophic constraints and volume continuity), we wanted to minimise unnecessary distance from the contour. Added a sentence in the text to justify this choice.

L160. It's not so surprising that EN4 and Argo agree closely as the Argo profiles are in EN4. Did you compare with a ship hydrographic section? Are the (complex) fronts and current meanders across this section captured in EN4?

There are no regular hydrographic sections across 47 N but as discussed in Section 2.4.5 we did compare with equivalent observations and model transects and found that the sub-1000 m geostrophic velocities calculated from EN4 data overestimated the strength of the Gulf Stream at

depth and underestimated the Deep Western Boundary Current and other southward flows across 47 N.  This was the main reason for requiring a conservation constraint to close the volume budget.  We have improved signposting to Section 2.4.5 to clarify our treatment of the EN4 data.

L174. A sensible constraint. What was the reference velocity and how much transport does it amount to in total? Please state in the paper.

We state the reference velocity and transport in Section 2.4.5 where this constraint is described in more detail.  Improved signposting to this section.

L178. The ADT requires an estimate of the geoid, which can be uncertain in the open ocean. How much do your results depend on the mean dynamic topography?

The ADT accounts for about 60 % of the variance for the heat and freshwater fluxes, but only 30 % of the variance for the overturning results.  Added this information in the 'estimation of uncertainties' paragraph, Section 2.4.2.  In general, we would hope that our results would be robust to local inaccuracies in the geoid because we accumulate flows over large horizontal scales.

L190, could I suggest using l or s instead of x for your along contour co-ordinate. X is very frequently used to mean zonal direction.

We have made it clearer that x is along-contour. As this coordinate system is used throughout the study (rather than switching back and forth with the zonal definition of x) we feel that it is reasonable to keep this notation.

L195, define Q, v in equation. Suggest using Qv to match later equations.

Defined Q and v as suggested.  The use of Q for volume flux is often used in similar studies and seems compatible with our notation in later equations, so have not changed to Qv.

L230, did the volume conservation constraint applied in the observations work in the Viking model?

As discussed in Section 2.4.5, the constraint was necessary in the observations because sub-1000 m geostrophic velocities calculated from EN4 data overestimated the strength of the Gulf Stream at depth and underestimated the Deep Western Boundary Current and other southward flows across 47 N when compared to dedicated observation campaigns and model studies.  This appears to be primarily due to data coverage and resolution limitations of EN4.  While model geostrophic velocities mimicked our methods by referencing the model sea surface, the corresponding property gradients at depth did not suffer from the same resolution problems.  The correction velocity required to balance the model geostrophic flows would therefore be smaller than that necessary for the observations.

L282, I don't find the overbar helpful notation

Changed to subscript "ref".

L295, counter-clockwise -> cyclonic

Text amended.

Fig4: fabulous figure. Please add colorbars.

Thank-you. Added colorbars to Fig. 4.

L297. I think 'negative' deserves more explanation: it means going to a higher density in a cyclonic direction?

Modified text to clarify.

Fig 5a. I'm not sure about arrows here. The arrows don't point in the direction of the current. They're constrained to be perpendicular to your section.

We feel that the arrows are helpful for contextualising the 2D transport figures but agree that they could give an incorrect impression of direction. Added the following text to relevant figures: "Quiver arrows show magnitude of geostrophic transport perpendicular to the section."

L346. Do you mean Goban Spur or the Porcupine Bank? It looks bigger than GS to me.

Agreed, changed to "Porcupine Bank".

L357. I'm struggling with export and a negative number in one line. 'Export of 12 Sv' or 'transport of -12Sv'?

Very unhelpful double negative: thank-you, amended.

Fig. 6a is hard to read the arrows. Really interesting breakdown of Ekman component. Why not the same colours for the geostrophic? Fig 5a?

Reduced the line width on the arrows on Figs. 5a and 6a to hopefully improve clarity. The Ekman component (Fig. 6a) has clear seasonality, so the magnitude of the arrows is distinct. By contrast we found that the geostrophic component (Fig. 5a) had little seasonality, and four arrows for each grid cell cluttered the plot without providing much insight.

Fig. 7. I like this a lot. Very convincing.

Thank-you for the comment.

Section 3.4. I need more context here. This overturning is different from say the OSNAP estimate as it's overturning around a closed contour around the subpolar gyre.

The phrase "overturning divergence" has been used by other studies to signpost this distinction between overturning within a closed contour and an open section such as OSNAP. While it doesn't strictly make sense to discuss the divergence of a non-vector quantity, it is useful phraseology. Rather than use this phrase throughout the paper, we have added a note to clarify the meaning of overturning in the context of this study.

Could you add the OSNAP mean to Fig. 9 for context? The overturning in this calculation occurs at a lighter density seems to be the key difference (OSNAP 27.5-27.7, here 27.3).

OSNAP mean is 14.9 Sv @ 27.66 kg/m3. As this is mentioned in the text (more prominently in response to Reviewer 1's comments) and is a different measure of overturning it doesn't seem necessary to adjust the axes away from the observations to make this comparison.

As this is a very OSNAP inspired paper—could you break the streamfunctions into an analogue of OSNAP east and OSNAP west?

A streamfunction for subsets of the boundary would not have the constraint of volume conservation so would be dominated by accumulation or loss of water driven by net volume flux rather than true overturning. We included 9b and c as they illustrated the contributions of the boundary vs. 47N transect, but we do not think that subsetting the OSNAP regions would be very informative.

Similarly, I would suggest adding OSNAP estimates of heat + fw flux to Fig. 10. You get half the heat flux and ¼ of the fwater flux of OSNAP.

This is an interesting suggestion, but we feel that this addition would detract from the figure. Our results are not strictly comparable to OSNAP because OSNAP Heat and FW fluxes are for the entire region north of the OSNAP line (i.e. all of the Arctic Ocean). Our results are effectively a divergence

of heat and FW within the boundary of our domain. If we were to make a direct comparison to OSNAP it would be a residual telling us what happens to the north of OSNAP / our northern boundary. However, as our northern boundary is not the OSNAP line it would not be a meaningful comparison.

L585. I don't agree that's what you're doing! Specifically you've calculate the flux across the 1000m isobath + 47 N. I think you need to say that you've built in a definition of interior and exterior at least.

Agreed, added "between the interior and exterior of the SPG" to clarify.

For the discussion, a visual summary would be very useful. It's hard to keep all the numbers in mind.

This was the motivation behind Fig. 12 in the discussion. We have improved signposting to Fig. 12 throughout the paper.

I like Fig. 12 and the discussion that goes with it.

Thank-you for the comment.

---

## Referee Report (RR1)

egusphere-2022-472

**Observation-based estimates of volume, heat and freshwater exchanges between the subpolar North Atlantic interior, its boundary currents and the atmosphere**
by Sam C. Jones, Neil J. Fraser, Stuart A. Cunningham, Alan D. Fox, and Mark E. Inall

Great to see the revised version of this manuscript. I think that it is much improved and will form a significant contribution to our understanding of overturning in the subpolar North Atlantic. I only have a few remarks.

**General comments:**

My first general comment was only partially addressed. I understand and accept the reasoning for keeping the perimeter contour at the 1000 m isobath. As a consequence, the contour is for large parts of the domain oriented along the major boundary current system (particularly in the Irminger Sea along east Greenland and in the Labrador Sea, e.g., Figure 2). This implies that the dominant isopycnal slope along this depth contour is across the boundary current system and that the cross-contour component of the flow is a relatively small residual relative to the much stronger along-contour flow. I did not register that this issue was discussed in the manuscript.

**Specific comments:**

Line 20:
I am surprised to see that there is no evidence of the substantial export of freshwater from the Arctic Ocean (e.g., Haine *et al.*, 2015) in the subpolar gyre budget, that you can close the budget considering only the atmospheric fluxes.

Line 27:
Perhaps rephrase, dense-water formation is not considered a "driver" of the AMOC (Kuhlbrodt *et al.*, 2007).

Line 54:
Lozier *et al.* (2019) did not distinguish south and north of the Greenland-Scotland Ridge, they only concluded that most of the overturning occurs east of Greenland.

Line 127:
I think that the statement "some profiles are used in more than one grid cell" is an understatement. If the minimal search radius is 150 km, which is the same as the distance between grid points, most of the profiles will be used in the two nearest grid cells. If you plot the search radius used for each grid cell on a map such as Figure 2a, I think this should be evident. Davis (1998) provides good justification for using a search radius that is reduced in the across-slope direction compared to the along-slope direction.

Line 299:
I think the statement that geostrophic flow is largely out of the subpolar gyre shallower than 500 db and opposite below 500 db should be elaborated on.

Lines 348 and 351:
Your estimates of the transport of Atlantic Water from the subpolar gyre into the Nordic Seas could be compared to the transport estimates from the monitoring efforts along the Greenland-Scotland Ridge (e.g., Østerhus *et al.*, 2019).

Line 421:

It appears that the values of 8 and 14 cm/s were simply estimated from Figure 2b. This could have been approached more quantitatively, for instance by making a frequency histogram of the speed between successive surfacings of individual floats within the basin perimeter.

Line 474:

East Greenland south of Denmark Strait is not a major overflow region. While the East Greenland Spill Jet may contribute some dense water to the lower limb of the AMOC (Pickart *et al.*, 2005), the Faroe Bank Channel/Iceland-Scotland Ridge and Denmark Strait are the only major overflow regions.

Line 521:

Please elaborate on why there is substantial densification in summer, when the air-sea heat fluxes are very low or even warming the ocean.

Line 540:

There is an apparent inconsistency with line 34 (heat transport of 0.31 PW). The 0.27 PW quoted here may be more appropriate when considering only the Atlantic Water component, but the difference with the previously quoted heat transport should probably be explained.

Line 631 and elsewhere:

The uncertainty estimate in the overturning strength is provided in the conclusion (line 836), it should also be included in the discussion section.

Line 675:

Please explain how the surface Ekman forcing introduces a lag in the overturning.

Line 714:

The Atlantic Water inflow from the subpolar gyre to the Nordic Seas east of Iceland is roughly evenly split on either side of the Faroe Islands (Østerhus *et al.*, 2019). Hence it would be more appropriate to ascribe this flow to the Iceland-Scotland Ridge than to the Faroe-Shetland Channel.

Line 818:

Note that the magnitude of the overflows east of Iceland (including entrainment) are of similar magnitude as the Denmark Strait overflow (Johns *et al.*, 2021). This should be reflected in the discussion, even if the model does not fully capture that component of the overflows from the Nordic Seas.

**Detailed comments:**

Line 30:

Arctic Ocean or Arctic Mediterranean would be more appropriate than Arctic (which by itself is ill-defined and typically also includes surrounding land masses) alone.

Line 394:

For clarity, perhaps specify that you mean the OSNAP **West** crossing.

Figure 12:

A sign is probably missing from the magnitude of the downward fluxes (all of the other transports have signs).

**References**

Davis RE. 1998. Preliminary results from directly measuring middepth circulation in the tropical and South Pacific. *Journal of Geophysical Research* **103**: 24 619–24 639, doi:10.1029/98JC01 913.

Haine TW, Curry B, Gerdes R, Hansen E, Karcher M, Lee C, Rudels B, Spreen G, de Steur L, Stewart KD, Woodgate R. 2015. Arctic freshwater export: Status, mechanisms, and prospects. *Global and Planetary Change* **125**: 13–35, doi:10.1016/j.gloplacha.2014.11.013.

Johns WE, Devana M, Houk A, Zou S. 2021. Moored observations of the Iceland-Scotland Overflow plume from along the eastern flank of the Reykjanes Ridge. *Journal of Geophysical Research: Oceans* **126**: doi:10.1029/2021JC017 524.

Kuhlbrodt T, Griesel A, Montoya M, Levermann A, Hofmann M, Rahmstorf S. 2007. On the driving processes of the Atlantic Meridional Overturning Circulation. *Reviews of Geophysics* **45**: RG2001, doi:10.1029/2004RG000 166.

Lozier MS, Li F, Bacon S, Bahr F, Bower AS, Cunningham SA, de Jong MF, de Steur L, de Young B, Fischer J, Gary SF, Greenan BJW, Holliday NP, Houk A, Houpert L, Inall ME, Johns WE, Johnson HL, Johnson C, Karstensen J, Koman G, Le Bras IA, Lin X, Mackay N, Marshall DP, Mercier H, Oltmanns M, Pickart RS, Ramsey AL, Rayner D, Straneo F, Thierry V, Torres DJ, Williams RG, Wilson C, Yang J, Yashayaev I, Zhao J. 2019. A sea change in our view of overturning in the subpolar North Atlantic. *Science* **363**: 516–521, doi:10.1126/science.aau6592.

Østerhus S, Woodgate R, Valdimarsson H, Turrell WR, de Steur L, Quadfasel D, Olsen SM, Moritz M, Lee CM, Larsen KMH, Jónsson S, Johnson C, Jochumsen K, Hansen B, Curry B, Cunningham S, Berx B. 2019. Arctic Mediterranean exchanges: A consistent volume budget and trends in transports from two decades of observations. *Ocean Science* **15**: 379–399, doi:10.5194/os–15–379–2019.

Pickart RS, Torres DJ, Fratantoni PS. 2005. The East Greenland Spill Jet. *Journal of Physical Oceanography* **35**: 1037–1053.

---

## Author Response (AR2)

Great to see the revised version of this manuscript. I think that it is much improved and will form a significant contribution to our understanding of overturning in the subpolar North Atlantic. I only have a few remarks.

**General comments:**

My first general comment was only partially addressed. I understand and accept the reasoning for keeping the perimeter contour at the 1000 m isobath. As a consequence, the contour is for large parts of the domain oriented along the major boundary current system (particularly in the Irminger Sea along east Greenland and in the Labrador Sea, e.g., Figure 2). This implies that the dominant isopycnal slope along this depth contour is across the boundary current system and that the cross-contour component of the flow is a relatively small residual relative to the much stronger along-contour flow. I did not register that this issue was discussed in the manuscript.

Thank-you for the additional suggestions for the manuscript. We agree that regions with a strong along-contour component to the flow will be particularly sensitive to temporal or spatial biases in the sampling. However, as we accumulate the geostrophic transports around the basin the distances over which the along-contour isopycnal slope is evaluated are large relative to the across-contour slope. Therefore, while along-contour flows may contaminate the signal for individual grid cells, they should have minimal impact on the accumulated transports. The uncertainties arising from our perturbation experiments provide some insight into the sensitivity of the results to local sampling errors (e.g. Fig. 5). Added text to clarify this point in the methods section of the manuscript [Line 200].

**Specific comments:**

Line 20: I am surprised to see that there is no evidence of the substantial export of freshwater from the Arctic Ocean (e.g., Haine *et al.*, 2015) in the subpolar gyre budget, that you can close the budget considering only the atmospheric fluxes.

The exported freshwater from the Arctic ocean enters and exits the SPG interior through the data curtain, so if the freshwater content is to remain in equilibrium, we would expect the advective flux across the data curtain to be approximately balanced by (precipitation - evaporation) over the SPG domain. In fact, Bryden et al. (2020) report a net loss of freshwater at a rate of 0.062 Sv for the region 26-70° N between 2008-2016 which would suggest less freshwater imported by surface fluxes than is exported through advective fluxes. As discussed after Line 738, we do see a small freshwater deficit, but it is within our error bounds. A proportion of the freshwater from the Arctic will pass around the perimeter of the basin in boundary currents and coastal currents without contributing to our budget.

Line 27: Perhaps rephrase, dense-water formation is not considered a "driver" of the AMOC (Kuhlbrodt *et al.*, 2007).

Agreed; replaced with "determinant" [Line 27].

Line 54: Lozier *et al.* (2019) did not distinguish south and north of the Greenland-Scotland Ridge, they only concluded that most of the overturning occurs east of Greenland.

The basins listed in brackets implied that we were only considering overturning south of the Greenland-Scotland Ridge, which was not our intention. We have removed these basins from the sentence which should make it clearer [Line 53].

Line 127: I think that the statement "some profiles are used in more than one grid cell" is an understatement. If the minimal search radius is 150 km, which is the same as the distance between grid points, most of the profiles will be used in the two nearest grid cells. If you plot the search radius used for each grid cell on a map such as Figure 2a, I think this should be evident. Davis (1998) provides good justification for using a search radius that is reduced in the across-slope direction compared to the along-slope direction.

Agreed; changed to "…most profiles are used in more than one grid cell." Thank-you for the reference: the point that decorrelation scales are greater in the along-slope direction is certainly relevant too. Added to text [Lines 124, 128].

Line 299: I think the statement that geostrophic flow is largely out of the subpolar gyre shallower than 500 db and opposite below 500 db should be elaborated on.

Agreed; this is a significant statement given the later overturning investigation. Added a sentence to expand on this observation [Line 310].

Lines 348 and 351: Your estimates of the transport of Atlantic Water from the subpolar gyre into the Nordic Seas could be compared to the transport estimates from the monitoring efforts along the Greenland-Scotland Ridge (e.g., Østerhus *et al.*, 2019).

Thank-you; added these comparisons to the text [Line 361].

Line 421: It appears that the values of 8 and 14 cm/s were simply estimated from Figure 2b. This could have been approached more quantitatively, for instance by making a frequency histogram of the speed between successive surfacings of individual floats within the basin perimeter.

We now compute float speeds using displacements between successive surfacings as suggested [Line 433 to 449]. This has resulted in a small increase in the estimated bottom Ekman transport (from 2.4 to 2.5 Sv), but this does not otherwise impact the results.

Line 474: East Greenland south of Denmark Strait is not a major overflow region. While the East Greenland Spill Jet may contribute some dense water to the lower limb of the AMOC (Pickart *et al.*, 2005), the Faroe Bank Channel = Iceland-Scotland Ridge and Denmark Strait are the only major overflow regions.

Agree that this statement was misleading; removed the reference to East Greenland [Line 498].

Line 521: Please elaborate on why there is substantial densification in summer, when the air-sea heat fluxes are very low or even warming the ocean.

The quantity we measure on the boundary is excess inflow at lighter densities and excess outflow of denser water. We interpret these flows as densification within the boundary (by surface fluxes). However, on seasonal timescales the inflow/outflow will be balanced by a combination of densification by surface fluxes and changing average density in the bounded volume. This density storage in the interior will result in a lag before modified water is registered at the boundary curtain, thus attenuating the seasonal cycle. Added this point in the overturning discussion [line 643].

Line 540: There is an apparent inconsistency with line 34 (heat transport of 0.31 PW). The 0.27 PW quoted here may be more appropriate when considering only the Atlantic Water component, but the difference with the previously quoted heat transport should probably be explained.

The more appropriate measure of heat transport is probably the 0.27 PW quoted at the second instance, as this is specific to the Greenland-Scotland Ridge. We have also attributed this figure to Chafik and Rossby, (2019) who are referenced by Tsubouchi et al., 2021 [Lines 34, 564].

Line 631 and elsewhere: The uncertainty estimate in the overturning strength is provided in the conclusion (line 836), it should also be included in the discussion section.

Added uncertainty estimates [Line 522, 721].

Line 675: Please explain how the surface Ekman forcing introduces a lag in the overturning.

Added an explanation of the mechanism by which surface Ekman influences the lag in overturning. Also clarified that, by removing Ekman forcing, the lag disappears in our results [Line 706]. As highlighted above, the seasonal cycle in overturning as measured by the flow across a section/boundary, is not constrained be in phase with the surface cooling cycle due to the storage term.

Line 714: The Atlantic Water inflow from the subpolar gyre to the Nordic Seas east of Iceland is roughly evenly split on either side of the Faroe Islands (Østerhus *et al.*, 2019). Hence it would be more appropriate to ascribe this flow to the Iceland-Scotland Ridge than to the Faroe-Shetland Channel.

Modified as suggested [Line 749].

Line 818: Note that the magnitude of the overflows east of Iceland (including entrainment) are of similar magnitude as the Denmark Strait overflow (Johns *et al.*, 2021). This should be reflected in the discussion, even if the model does not fully capture that component of the overflows from the Nordic Seas.

Thank-you for highlighting this. We have modified the discussion to reflect the comparable magnitude of the overflows. We have also stressed that VIKING20X may underestimate the Faroe Bank Channel overflow [Line 763, 855, 859, 866, 870].

**Detailed comments:**

Line 30: Arctic Ocean or Arctic Mediterranean would be more appropriate than Arctic (which by itself is ill-defined and typically also includes surrounding land masses) alone.

Text modified [Line 30].

Line 394: For clarity, perhaps specify that you mean the OSNAP West crossing.

Text modified [Line 408].

Figure 12: A sign is probably missing from the magnitude of the downward fluxes (all of the other transports have signs).

Modified Fig. 12 as suggested.

**References**

Davis RE. 1998. Preliminary results from directly measuring mid-depth circulation in the tropical and South Pacific. *Journal of Geophysical Research* 103: 24 619–24 639, doi:10.1029/98JC01 913.

Haine TW, Curry B, Gerdes R, Hansen E, Karcher M, Lee C, Rudels B, Spreen G, de Steur L, Stewart KD, Woodgate R. 2015. Arctic freshwater export: Status, mechanisms, and prospects. *Global and Planetary Change* 125: 13–35, doi:10.1016/j.gloplacha.2014.11.013.

Johns WE, Devana M, Houk A, Zou S. 2021. Moored observations of the Iceland-Scotland Overflow plume from along the eastern flank of the Reykjanes Ridge. *Journal of Geophysical Research: Oceans* 126: doi:10.1029/2021JC017 524.

Kuhlbrodt T, Griesel A, Montoya M, Levermann A, Hofmann M, Rahmstorf S. 2007. On the driving processes of the Atlantic Meridional Overturning Circulation. *Reviews of Geophysics* 45: RG2001, doi:10.1029/2004RG000 166.

Lozier MS, Li F, Bacon S, Bahr F, Bower AS, Cunningham SA, de Jong MF, de Steur L, de Young B, Fischer J, Gary SF, Greenan BJW, Holliday NP, Houk A, Houpert L, Inall ME, Johns WE, Johnson HL, Johnson C, Karstensen J, Koman G, Le Bras IA, Lin X, Mackay N, Marshall DP, Mercier H, Oltmanns M, Pickart RS, Ramsey AL, Rayner D, Straneo F, Thierry V, Torres DJ, Williams RG, Wilson C, Yang J, Yashayaev I, Zhao J. 2019. A sea change in our view of overturning in the subpolar North Atlantic. *Science* 363: 516–521, doi:10.1126/science.aau6592.

Østerhus S, Woodgate R, Valdimarsson H, Turrell WR, de Steur L, Quadfasel D, Olsen SM, Moritz M, Lee CM, Larsen KMH, J´onsson S, Johnson C, Jochumsen K, Hansen B, Curry B, Cunningham S, Berx B. 2019. Arctic Mediterranean exchanges: A consistent volume budget and trends in transports from two decades of observations. *Ocean Science* 15: 379–399, doi:10.5194/os–15–379–2019.

Pickart RS, Torres DJ, Fratantoni PS. 2005. The East Greenland Spill Jet. *Journal of Physical Oceanography* 35: 1037–1053.

**Additional References**

Bryden, H.L., Johns, W.E., King, B.A., McCarthy, G., McDonagh, E.L., Moat, B.I. and Smeed, D.A.: Reduction in ocean heat transport at 26° N since 2008 cools the eastern subpolar gyre of the North Atlantic Ocean. J. Clim., 33(5), 1677-1689, doi:10.5194/os-16-863-2020, 2020.

---

## Author Response (AR3)

**Author's response to editor's comments**

As far as I can tell, you have not responded to Reviewer 2's suggestion to include in the manuscript some of the material discussing the ANDRO product that you put in the responses to reviewers? Apologies if I missed this. It would be good to include brief mention of this to help your future readers?

Thank-you for the suggestion. We have added notes in the geostrophic transports section [line 206] and in the bottom Ekman section [line 435] explaining why we didn't consider the ANDRO product to be suitable for our analysis.